# Multi-angular polarimetric remote sensing to pinpoint global aerosol absorption and direct radiative forcing

Cheng Chen [1,2] ✉, Oleg Dubovik [1] ✉, Gregory L. Schuster[3], Mian Chin [4], Daven K. Henze [5], Tatyana Lapyonok[1], Zhengqiang Li[6], Yevgeny Derimian[1] & Ying Zhang[6]

Quantitative estimations of atmospheric aerosol absorption are rather uncertain due to the lack of reliable information about the global distribution. Because the information about aerosol properties is commonly provided by single-viewing photometric satellite sensors that are not sensitive to aerosol absorption. Consequently, the uncertainty in aerosol radiative forcing remains one of the largest in the Assessment Reports of the Intergovernmental Panel on Climate Change (IPCC AR5 and AR6). Here, we use multi-angular polarimeters (MAP) to provide constraints on emission of absorbing aerosol species and estimate global aerosol absorption optical depth (AAOD) and its climate effect. Our estimate of modern-era mid-visible AAOD is 0.0070 that is higher than IPCC by a factor of 1.3-1.8. The black carbon instantaneous direct radiative forcing (BC DRF) is +0.33 W/m$^2$ [+0.17, +0.54]. The MAP constraint narrows the 95% confidence interval of BC DRF by a factor of 2 and boosts confidence in its spatial distribution.

Atmospheric aerosols scatter and absorb solar and terrestrial radiation, thereby cooling and warming the atmosphere-earth system. Current estimates of the global aerosol radiative effect indicate that aerosols have a net cooling effect on our planet, which partially offsets warming effects by greenhouse gases[1,2]. As a result of the limited understanding of details in the global distribution of aerosol absorption, global climate effects by atmospheric aerosols remain one of the largest forcing uncertainties in the 5th and the recent 6th Intergovernmental Panel on Climate Change (IPCC AR5 and AR6) assessments[1,3]. Although the aerosol absorption by black carbon (BC) aerosol is known to be one of the largest contributors with carbon dioxide ($CO_2$) and methane ($CH_4$) for heating our planet[4–6], there are still significant challenges for pinpointing the effects of absorbing aerosols[7]. The recent AR6[8] reports -50% reduction of BC warming effects by adjusting its rapid climate responses. Nonetheless, the heterogeneity of global spatial distribution of aerosol absorption has also

certainly an impact on the rapid adjustments. Thus, the improved quantification of global spatial heterogeneity of aerosol absorption distribution is still highly demanded. Indeed, the climate models produce a large diversity in simulations of global aerosol absorption[9–12], largely because of the scarceness of reliable global long-term observation of aerosol absorption to constrain the models. At present, the aerosol absorption optical depth (AAOD) retrieved from sun-sky measurements at worldwide Aerosol Robotic Network (AERONET) stations[13,14] is the main product used to evaluate and constrain climate models[15–20], and most models underestimate aerosol absorption significantly when compared to AERONET AAOD[10,11]. However, the AERONET-derived aerosol single scatter albedo (SSA), which is the ratio of scattering to total extinction (SSA = 1-AAOD/AOD), is of high uncertainty at low aerosol abundance[21,22]. Therefore, the highest quality AERONET Level 2 inversion products are provided only when the aerosol optical depth (AOD) at the blue channel is higher than 0.4.

[1]Univ. Lille, CNRS, UMR 8518 - LOA - Laboratoire d'Optique Atmosphérique, F-59000 Lille, France. [2]GRASP-SAS, Univ. Lille, Villeneuve d'Ascq 59650, France. [3]NASA Langley Research Center, Hampton, VA 23681, USA. [4]NASA Goddard Space Flight Center, Greenbelt, MD 20771, USA. [5]Department of Mechanical Engineering, University of Colorado, Boulder, CO 80309, USA. [6]Aerospace Information Research Institute, Chinese Academy of Sciences, Beijing 100101, China. ✉e-mail: cheng.chen@grasp-sas.com; oleg.dubovik@univ-lille.fr

The high uncertainty in the AERONET SSA at low AOD and the Level 2 products mainly for moderate and high AOD conditions could be contributing a significant bias to the model simulations and AERONET intercomparison[23].

Generally, most climate models simulate aerosol absorption that is notably weaker compared to the values derived (inverted) from remote sensing measurements such as AERONET. There are two potential reasons. First, some of the key mechanisms in the global climate models (GCMs) that could enhance aerosol absorption are not rigorously implemented or simply missing. This includes, for example, aspects such as internal mixing of BC with other species[24,25], accounting for particle morphology[26], BC lofted to high altitudes[27], and the limited spatial resolution of model and emission inventories;[28] all of these issues are very difficult to include comprehensively in the models. Second, the lack of long-term global large-scale aerosol absorption measurements makes it very difficult to constrain and validate GCM simulations. Correspondingly, the model estimates of aerosol absorption effects on climate are quite diverse. For example, Ramanathan and Carmichael[29] reported a global BC instantaneous radiative forcing of +0.9 W/m², with an uncertainty range of +0.4 to +1.2 W/m², which is as much as ~50% of the magnitude of $CO_2$ forcing. The Aerosol Comparisons between Observations and Models (Aero-Com) Phase II multi-model experiment estimated a much smaller value of +0.23 W/m² (Myhre et al.[30]). Bond et al.[9] utilized AERONET AAOD products to scale global model modern-era anthropogenic BC AAOD to 0.0049, and suggested that the BC radiative forcing is +0.51 [+0.06, +0.91] W/m². IPCC AR5 adopted some expert judgements and reported the estimate +0.4 W/m² which is halfway between AeroCom Phase II (Myhre et al.[30]) and Bond et al.[9] with an uncertainty range [+0.05, +0.8] W/m². Overall, GCMs struggle to realistically represent global aerosol absorption; thus, the modeled BC radiative forcing is diverse and of low confidence.

The above studies suggest that the ground-based AERONET point measurements alone are not fully sufficient for constraining aerosol simulations, and reliable global satellite data of aerosol absorption are desirable to assess global aerosol absorption. Nevertheless, the commonly used single-viewing photometric satellite sensors, such as AVHRR, MODIS, MERIS, VIIRS etc., which have provided continuous and valuable information to understand global AOD and its climate effects over the past two decades[31–34], have weak or no sensitivity to aerosol absorption. In this regard, the scientific community anticipates reliable observations of global aerosol absorption from multi-angular polarimetry (MAP)[35,36], which has a high potential for characterizing aerosol properties, including absorption, size, and sources[37–40]. Recently, some enhanced aerosol retrieval algorithms have been developed for the practical utilization of the extensive information content from MAP measurements[41,42]. Schutgens et al.[43] reported convincing consistency in available satellite aerosol absorption data, but some non-negligible discrepancies between different satellite products remain to be understood.

This study was motivated by the efforts of Dubovik et al.[41] and Chen et al.[44], which provide a long-term record of global spectral aerosol absorption products from the MAP measurements of POLDER using the GRASP (Generalized Retrieval of Atmosphere and Surface Properties) algorithm (www.grasp-open.com). POLDER (Polarization and Directionality of the Earth's Reflectances) was developed to measure spectral directional polarized solar radiance reflected by the Earth-atmosphere system[45]. POLDER-3 instrument on board PARASOL satellite was the longest to date operational space-borne MAP sensor, while POLDER-1 and 2 have rather limited time series of measurements[38,40]. GRASP is a highly capable and versatile remote sensing retrieval algorithm[46,47] that has evolved from an earlier implementation on operational aerosol retrievals using AERONET radiometers[14]. This algorithm was used to generate a global POLDER-3/GRASP record of the spectral AAOD that agrees to within ±0.01 of the

values in the AERONET database, covering wavelengths from the shortwave visible to the near-infrared (VIS-NIR)[44]. Furthermore, the POLDER-3/GRASP products were used to constrain global aerosol emissions of three absorbing aerosol species (BC, organic aerosol - OA, and desert dust - DD) by direct fitting of the observed spectral AAOD using the GEOS-Chem transport model[48,49]. Thus, in this paper, we use these MAP-constrained absorbing aerosol emission database (MACE) to demonstrate that they are useful for improving the model simulation of aerosol absorption and could be further used to pinpoint aerosol climate effects due to absorption.

## Results

### Observationally constrained modeled aerosol absorption

We employ the aerosol simulation using GEOS-Chem v11-01 (Heald et al.[50]). Specifically, five externally mixed aerosol components: BC, OA, DD, sulfate-nitrate-ammonium (SNA), and sea salt (SS) are simulated. We follow the similar GEOS-Chem scheme (meteorology, physical and chemical processes) as Heald et al.[50] where an early version v9-01 was used. Moreover, by replacing the a priori emission inventories with MAP-constrained 6 years (2006–2011) BC, OA, and DD daily emissions from MACE, we obtain the observationally constrained aerosol absorption simulation (see the Methods section). Meanwhile, we conduct two simulations using (i) only natural emissions and (ii) modern-era total (natural + anthropogenic) emissions. By subtracting the natural contribution from the total, i.e. (ii)−(i), we assess the contribution and effects due to the anthropogenic emissions. The separation of natural/anthropogenic emissions in MACE database is based on the daily proportion in the GEOS-Chem model over each grid box (Methods), where the fossil fuel and biofuel emissions are the anthropogenic sources of BC and OA, and the biomass-burning emissions are categorized as natural sources. In addition, 20% of dust emission are defined as anthropogenic origins (Heald et al.[50]).

Figure 1 shows the global distribution of aerosol absorption simulated based on the MACE database. The simulated daily AAOD were averaged from 2006 to 2011 to obtain the modern-era global spatial distribution of AAOD at 550 nm (mid-visible) shown in Fig. 1a. As can be seen, the high aerosol absorption is observed over industrial and biomass-burning regions. The globally averaged AAOD is estimated at 0.0070 in the mid-visible spectrum, with the 95% confidence interval [0.0068, 0.0073] inferred as $2\sigma$ based upon 6 years of simulations. The three major contributors to the modern-era value of AAOD (0.0070) are BC (0.0055), OA (0.0007), and DD (0.0008). Because of the observationally constraint on spectral, spatial, and temporal variability of aerosol absorption, we would put high confidence on our estimation of modern-era global aerosol absorption as well as its speciation. The confidence of pinpoint modern-era global aerosol absorption is confirmed by 2 independent assessments using observational constraints from AERONET: (i) the modern-era BC AAOD 0.0055 [0.0052, 0.0057] is close to the value from Bond et al.[9] of 0.0061, which scaled model simulations to AERONET AAODs; (ii) the constrained modern-era AAOD 0.0070 [0.0068, 0.0073] is consistent with the value of 0.0072 from the Max Planck Institute Climatology Version 2 (MACv2)[51] obtained by constraining AeroCom models using AERONET climatology. Meanwhile, in comparison with typical modeling study from the AeroCom Phase II modern-era AAOD 0.0042 ($1\sigma = 0.0019$)[30] and the recent Phase III AAOD 0.0054 ($1\sigma = 0.0023$), BC AAOD 0.0030 [0.0007, 0.0077][12,52], our estimate still suggests a ~1.3–1.8 times stronger global aerosol absorption than the multi-model assessments, which is outside the $1\sigma$ range yet within the maxima of the Phase II experiment, and it is within the $1\sigma$ range of the Phase III experiment. However, the 95% confidence intervals of our estimates are pinpointed significantly (Fig. 1b). Figure 1c shows the modern-era global distribution of anthropogenic AAOD fraction, which corresponds to the AAOD fractional value due to anthropogenic emissions. Generally, the Northern Hemisphere (NH) has a higher anthropogenic

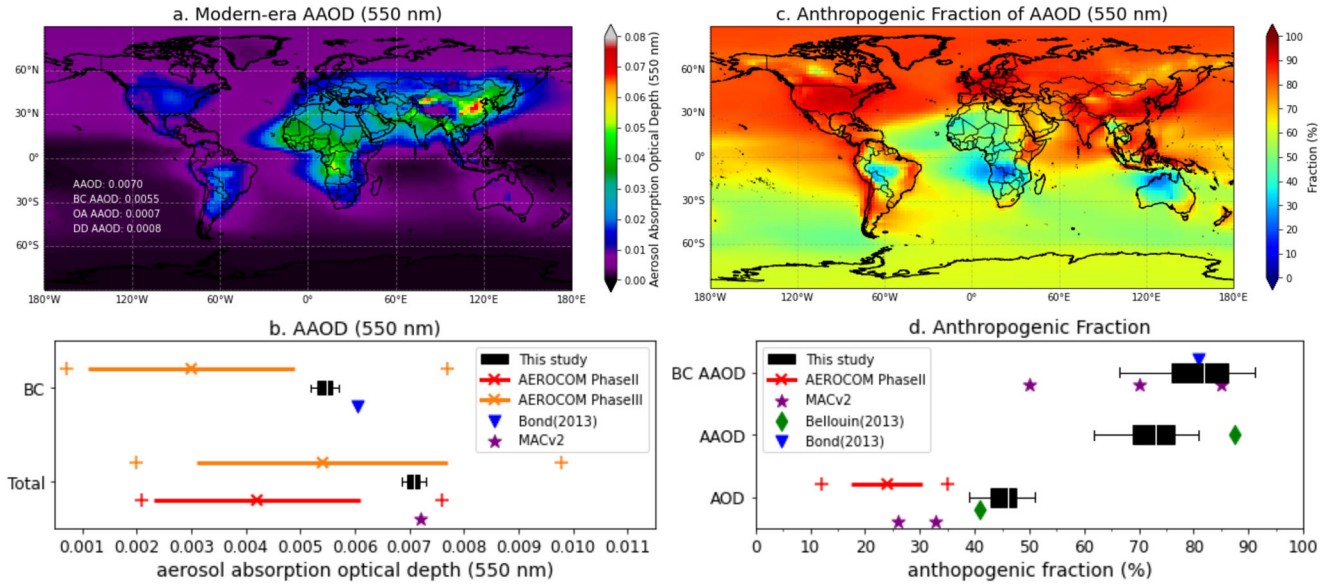

**Fig. 1 | Global distribution of aerosol absorption constrained by MAP satellite observations. a** Modern-era global aerosol absorption optical depth (AAOD) at 550 nm averaged from 2006 to 2011. **b** Estimation of global mean total aerosol absorption and its contribution from the primary aerosol absorber BC and inter-comparison with the Aerosol Comparisons between Observations and Models (AeroCom) Phase II and Phase III, Bond[9], the Max Planck Institute Climatology Version 2 (MACv2)[51]. **c** Global distribution of anthropogenic fraction of AAOD.

**d** Estimation of global mean anthropogenic fraction of AOD, AAOD, and black carbon (BC) AAOD and intercomparison with AeroCom Phase II, Bond[9], MACv2[51], and Bellouin[53]. The boxplots represent all mean estimations, 25th and 75th quantiles, and the 95% confidence intervals. For AeroCom Phase II and Phase III, the line indicates the mean ±1σ and the minimum and maximum values from individual models are also shown as crosses.

fraction of aerosol absorption than the Southern Hemisphere (SH), and the map also indicates that the AAOD over the Arctic region is largely (>85%) from anthropogenic origins. Meanwhile, the regions strongly affected by biomass burning and desert dust have relatively low anthropogenic fractions.

Although our constrained emissions technique indicates that the total value of AAOD = 0.0070 is of high confidence, the estimation of the contribution from anthropogenic activities relies on the separation of anthropogenic and natural emissions (see the Methods section), which cannot be inferred from satellite observations of total (anthropogenic + natural) column without additional a priori information. Therefore, medium confidence is assigned for our modern-era anthropogenic component of AAOD 0.0051 [0.0030, 0.0070], consisting of BC 0.0044 [0.0025, 0.0062], OA 0.0005 [0.0003, 0.0007], and DD 0.0002 [<0.0001, 0.0003]. The 95% confidence interval is derived from monthly simulations from 2006 to 2011. As shown in Fig. 1d, the global mean anthropogenic fraction of AAOD is 72.9% with 95% confidence interval of [58.2% to 79.8%]. Note that the anthropogenic fraction of AAOD is highly uncertain and inconclusive in AR5 (Boucher et al.[1]). Bellouin et al.[53] estimated a bit higher value than our constrained technique, around 87.5%, partially due to their consideration of biomass burning as being 100% anthropogenic. Our estimate of anthropogenic fraction for BC AAOD is 80.3% [65.9%, 88.9%], which is consistent with Bond et al.[9] (80.8%) and slightly higher than MACv2 (Kinne et al.[51]) perturbation range from 50% to 85%. Meanwhile, we report the modern-era anthropogenic fraction of mid-visible AOD 45.3% [37.9%, 50.5%], which is in line with Bellouin et al.[53] (~41%) and higher than AeroCom Phase II (24%, 1σ=6.5%), MACv2 (~26% to 33%), and also higher than AR5 [20%, 40%]. In general, the anthropogenic fraction of AAOD is at least ~1.6 times higher than the anthropogenic fraction of AOD. In addition, the anthropogenic fraction of BC AAOD (80.3%) is even higher than that of AAOD (72.9%). Hence, this suggests that elevated aerosol absorption is largely affected by human activities and is among the crucial factors of pollution-induced climate change effects.

## Estimation of aerosol radiative effects due to its absorption

The perturbation of instantaneous solar and terrestrial radiation by the presence of aerosols in the atmosphere defines aerosol direct radiative effects (DRE). The DRE caused by anthropogenic aerosol emitted during the industrial era is known as the instantaneous aerosol direct radiative forcing (DRF), and is commonly used to quantify anthropogenic aerosol effects on the global temperature and climate system[54–56]. The effective radiative forcing also includes rapid adjustments to the atmosphere and surface[57], and this has become the more commonly used indicator of merit because of its accurate measure of the Earth's radiative imbalance due to particular climate forcing agents and better indication of forcing on global temperature change[3,58]. However, the accurate quantification of the effects caused by the rapid adjustments relies on the knowledge of source and spatial distribution, such as aerosol shortwave absorption. In this study, we still refer to instantaneous DRF unless otherwise specified. Previous studies show a best estimate of modern-era aerosol DRF at the level of ~−0.3 W/m² (Myhre et al.[30,59], Thornhill et al.[58], Kinne et al.[51]) with a large uncertainty range that includes positive warming effects. A recent review by Bellouin et al.[2] reported aerosol DRF ranging from −0.05 to −0.45 W/m² as 95% confidence intervals, and it was adopted by IPCC AR6 with a best estimate −0.25 W/m² (1σ = 0.2)[3].

In this study, we run GEOS-Chem (v11-01) coupled with a radiative transfer mode (RRTMG) to further calculate the aerosol radiative effects (Heald et al.[50]). Similarly, the emissions of absorbing aerosol species (BC, OA, and DD) are updated from the MACE database. Simulations conducted with and without anthropogenic emissions are used to derive the aerosol DRF (DRE caused by the aerosol emitted from anthropogenic origins). Even though the observational constraint is only implemented on three absorbing aerosol species, we still assess the total aerosol effects by simulation of scattering aerosol SNA and SS simultaneously based on the standard scheme in GEOS-Chem model. The previous estimation by Heald et al.[50] using the same model with an early version showed good correspondence with AeroCom Phase II, where the DRF for SNA is ~−0.4 W/m². Hence, the changes in

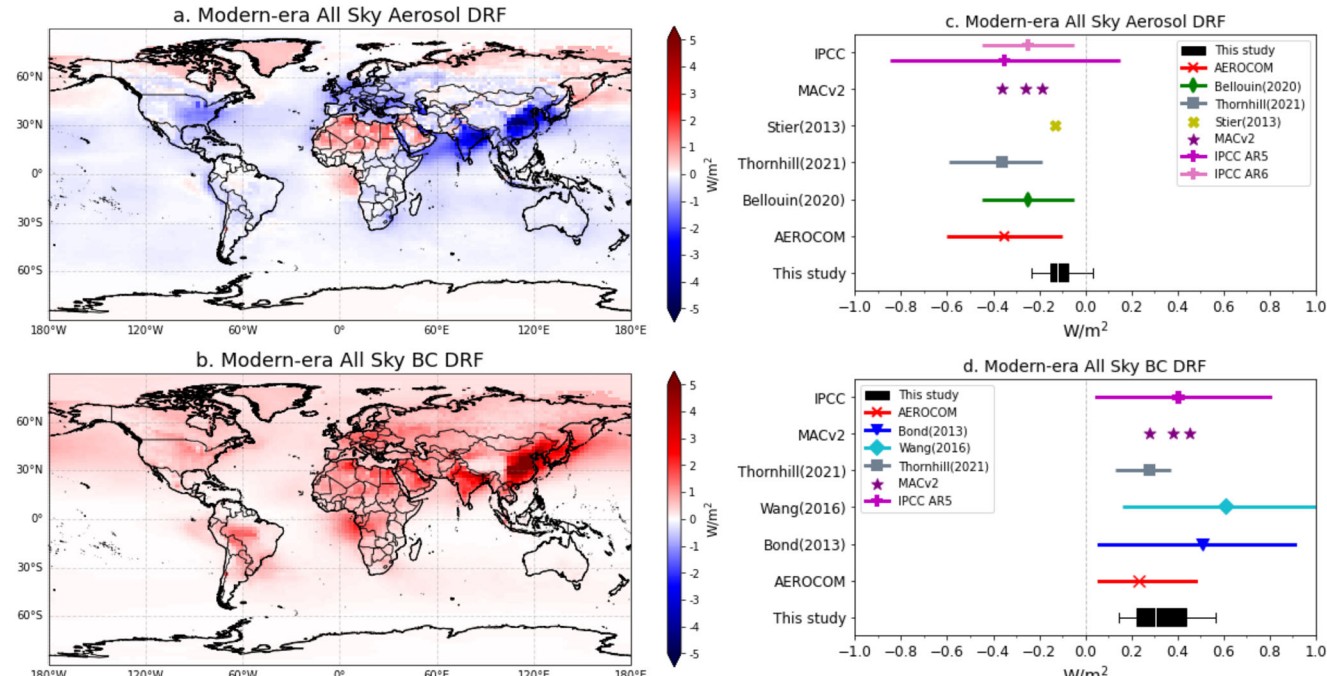

**Fig. 2 | Modern-era aerosol direct radiative forcing (DRF) and black carbon (BC) DRF constrained by satellite observations. a** Spatial distribution of modern-era all-sky aerosol DRF averaged from 2006 to 2011. **b** Spatial distribution of modern-era all-sky BC DRF averaged from 2006 to 2011. **c** Modern-era aerosol DRF inter-comparison with the Assessment Reports of the Intergovernmental Panel on Climate Change (IPCC AR5 and AR6), Bellouin[2], Stier[83], the Max Planck Institute

Climatology Version 2 (MACv2)[51], the Coupled Model Intercomparison (CMIP6)[58], and the Aerosol Comparisons between Observations and Models (AeroCom) Phase II. **d** Modern-era BC DRF intercomparison with AR5, Bond[9], Wang[84], MACv2, CMIP6 and AeroCom Phase II. The boxplots represent all mean estimations, 25th and 75th quantiles, and the 95% confidence intervals.

DRF from our estimation are mainly driven by constrained emissions of absorbing aerosols. The cross-comparison of the simulations using a priori emission (Heald et al.[50]) and MAP-constrained MACE emission based on the GEOS-Chem RRTMG model is present in Supplementary Table 1. Figure 2 shows the modern-era aerosol and primary absorber BC DRF at the top-of-atmosphere (TOA) based on all-sky conditions. As shown in Fig. 2a, the total aerosol radiative effect is cooling over industrial regions, such as China, India, the US, and Europe. A warming effect is also evident in the Arctic region, where the transported absorbing aerosol is dominant. The dust absorption in the longwave results in warming effects over the Sahara. The southeast Atlantic, where the absorbing aerosol interacts with subtropical stratocumulus, is another region where the aerosol warming effect was previously underestimated by climate models[60].

Our best estimate of modern-era aerosol global DRF is −0.14 W/m² (1$\sigma$ = 0.08 W/m²) with a 95% confidence interval of [−0.25, +0.01] W/m², based upon 6 years of simulations from 2006 to 2011. The 95% confidence interval is estimated by fixing the global annual mean aerosol extinction and absorption while perturbing anthropogenic fraction of absorbing aerosol from monthly simulations. Our aerosol DRF uncertainty range is driven by single model 6 years variability and perturbation of BC anthropogenic fractions. Many other factors, for example uncertainties of the model itself, the uncertainties in parameterization of chemical and physical processes, meteorology, limited spatial resolution of model simulation, etc., were not taken into account in the present estimation. Basically, our estimation is different with the AeroCom, CMIP6[58] (Coupled Model Intercomparison), AR5, and AR6 multi-model assessments, as shown in Fig. 2c. The IPCC AR5 DRF is −0.35 W/m² (1$\sigma$ = 0.5 W/m²), AeroCom Phase II DRF is −0.35 W/m² (1$\sigma$ = 0.15 W/m²), Thornhill et al.[58] obtained −0.36 W/m² [−0.19, −0.49] using CMIP6 models, AR6 and Bellouin et al.[2] reported −0.25 W/m² (1$\sigma$ = 0.2 W/m²). Thus, our estimate indicates 45-60% greater warming and a 95% confidence interval that narrows to [−0.25, +0.01], with 2–4

times smaller than the other simulations. It should be noted that the estimations of uncertainty range are different from study to study. On the other hand, our estimate is consistent with the DRF of −0.13 W/m² that Stier et al.[61] obtained using the comprehensive aerosol absorption calculation in the ECHAM5-HAM model, and is close to MACv2[51] reported −0.19 W/m² by adjusting BC anthropogenic fraction to 85%.

Figure 2b shows the spatial distribution of modern-era DRF due to primary aerosol absorber BC (BC DRF). The BC effect is globally warming except for some regions nearly neutral effect dominated by biomass burning (e.g., South America), which is categorized as a natural source in this study. As shown in Fig. 2d, our best estimate of modern-era BC DRF is +0.33 W/m² with a 95% confidence interval [+0.17, +0.54] W/m² and 1$\sigma$ equals 0.13 W/m². Heald et al.[50] used the bottom-up emission inventories in the same GEOS-Chem RRTMG simulation, which resulted in a BC DRF of +0.078 W/m². The use of MACE BC emissions leads to an increase of our simulated BC DRF of about a factor of 4. Additionally, the overall aerosol DRF increases ~60% from −0.36 W/m² (Heald et al.[50]) to −0.14 W/m², which is largely explained by the underestimation of BC radiative effects using the bottom-up emissions without constraints from observations. More detailed intercomparison with the simulation based on a priori emissions in the same model can be found in Methods.

Meanwhile, the estimated +0.33 W/m² BC DRF is in line with the IPCC AR5 + 0.4 W/m² that is roughly halfway between AeroCom Phase II (+0.23 W/m²) and Bond (2013)[9] (+0.51 W/m²) (Fig. 2d), although our 95% confidence interval is significantly narrowed owing to the observational constraints. Even though our estimation of modern-era BC AAOD is 10% smaller than Bond[9], which scaled the model AAOD using ground-based AERONET measurements, the satellite products provide additional details in aerosol spatial distribution that are non-negligible for radiative effect estimates[62] and hardly seen from the ground-based network. Additionally, the radiative forcing efficiency varies spatially and temporally. For example, in our simulation, the BC forcing

efficiency is much smaller (~40–70 W/m² per unit AAOD) over industrial regions than the other areas (105 W/m² per unit AAOD) (Table 1). As a consequence, the BC DRF discrepancy (a factor of ~1.5) for the two approaches (+0.33 vs. +0.51 W/m²) is greater than global anthropogenic BC absorption (0.0044 vs. 0.0049) discrepancy (a factor of ~1.1). Therefore, the spatial distribution of aerosol absorption obtained from satellite remote sensing (i.e., MAP) is crucial for adequate constraining of the aerosol climate effects.

We note that the main contributor of anthropogenic AAOD (0.0051) is BC (accounting for ~86% of anthropogenic aerosol absorption), followed by OA (10%) and DD (4%). Figure 3a shows the BC contribution to anthropogenic aerosol absorption is over 80% at all

**Table 1 | Summary of regional anthropogenic black carbon (BC) aerosol absorption optical depth (AAOD) at 550 nm, BC direct radiative forcing (DRF), and BC DRF efficiency based on our simulation**

| Regions | Anthropogenic BC AAOD | BC DRF efficiency (W/m² per AAOD) | BC DRF (W/m²) |
|---|---|---|---|
| ESA | 0.0183 | 57.4 | 1.05 |
| SAS | 0.0176 | 43.2 | 0.76 |
| NAM | 0.0092 | 39.1 | 0.36 |
| EUR | 0.0099 | 50.5 | 0.50 |
| SAM | 0.0070 | 44.3 | 0.31 |
| CAF | 0.0172 | 69.2 | 1.19 |
| Rest of world | 0.0019 | 105.3 | 0.20 |

The six selected regions are East Asia (ESA), South Asia (SAS), North America (NAM), Europe (EUR), South America (SAM), and Central Africa (CAF), respectively.

latitudes, and it is relatively higher in the Northern Hemisphere (NH) than in the Southern Hemisphere (SH). Moreover, the highest anthropogenic AAOD and BC AAOD are shown at NH middle latitudes, while the highest BC AAOD ratio is observed at NH high latitudes. Figure 3b shows the latitude-dependent aerosol DRF (no BC) and the BC DRF. In general, the spread is much larger in the NH than in the SH, and the BC direct effect is warming (positive DRF) at all latitudes. The peak BC warming effect (~+1.0 W/m²) is observed at the NH middle latitudes where the BC AAOD is also the highest. Meanwhile, the highest zonal BC DRF at NH middle latitudes is ~+0.4 w/m² in the AeroCom Phase II experiment[30], which is about 2.5 times smaller than our estimates. Additionally, no individual model reaches the peak level of +1.0 W/m² shown in Fig. 3b, with the closest value of ~+0.8 w/m² reported by the CAM4-Oslo model[30]. Basically, the warming effect due to the presence of BC offsets about 2/3 of the cooling effect caused by other aerosol species (i.e., aerosols without BC) at all latitudes, especially at the NH middle latitudes where anthropogenic activities and the observed BC warming effect are high.

**Regional trends of anthropogenic BC emission, AAOD, and DRF**
In order to analyze explicitly the effects over the industry regions, we select six main industrial regions covering the same ranges in longitude and latitude (center coordinates ±30° longitude, ±22° latitude): East Asia (ESA: 130°E, 32°N), South Asia (SAS: 70°E, 27°N), North America (NAM: 95°W, 37°N), Europe (EUR: 50°W, 52°N), South America (SAM: 60°W, 18°S) and Central Africa (CAF: 10°E, 3°S). The areas of 6 regions are drawn in Fig. 4a. In addition, Fig. 4a shows the spatial distribution of annual mean anthropogenic BC emissions from 2006 to 2011. It is worth noting that the emissions have significantly changed after 2011 over East Asia;[63] for example, a significant decreasing trend is

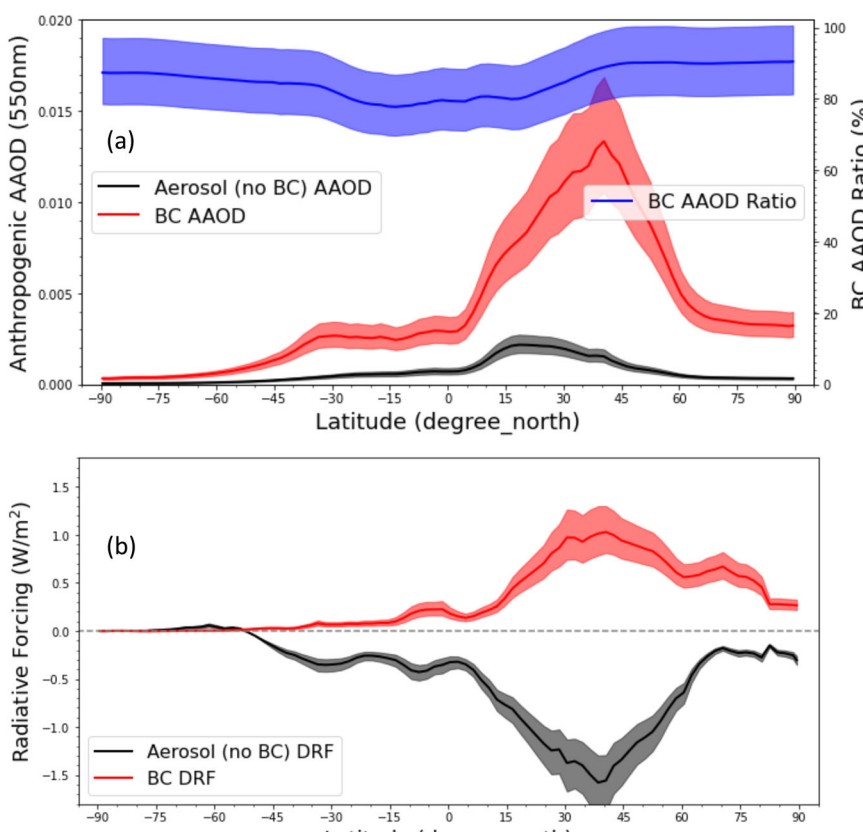

**Fig. 3 | Latitude-dependent anthropogenic aerosol absorption and direct radiative forcing. a** Zonal mean anthropogenic aerosol (no BC), BC absorption, and the fraction of anthropogenic aerosol absorption optical depth (AAOD) associated with black carbon (BC). **b** Zonal mean all-sky aerosol (no BC) DRF and BC direct radiative forcing (DRF). Shaded area represents the 95% confidence intervals of the mean.

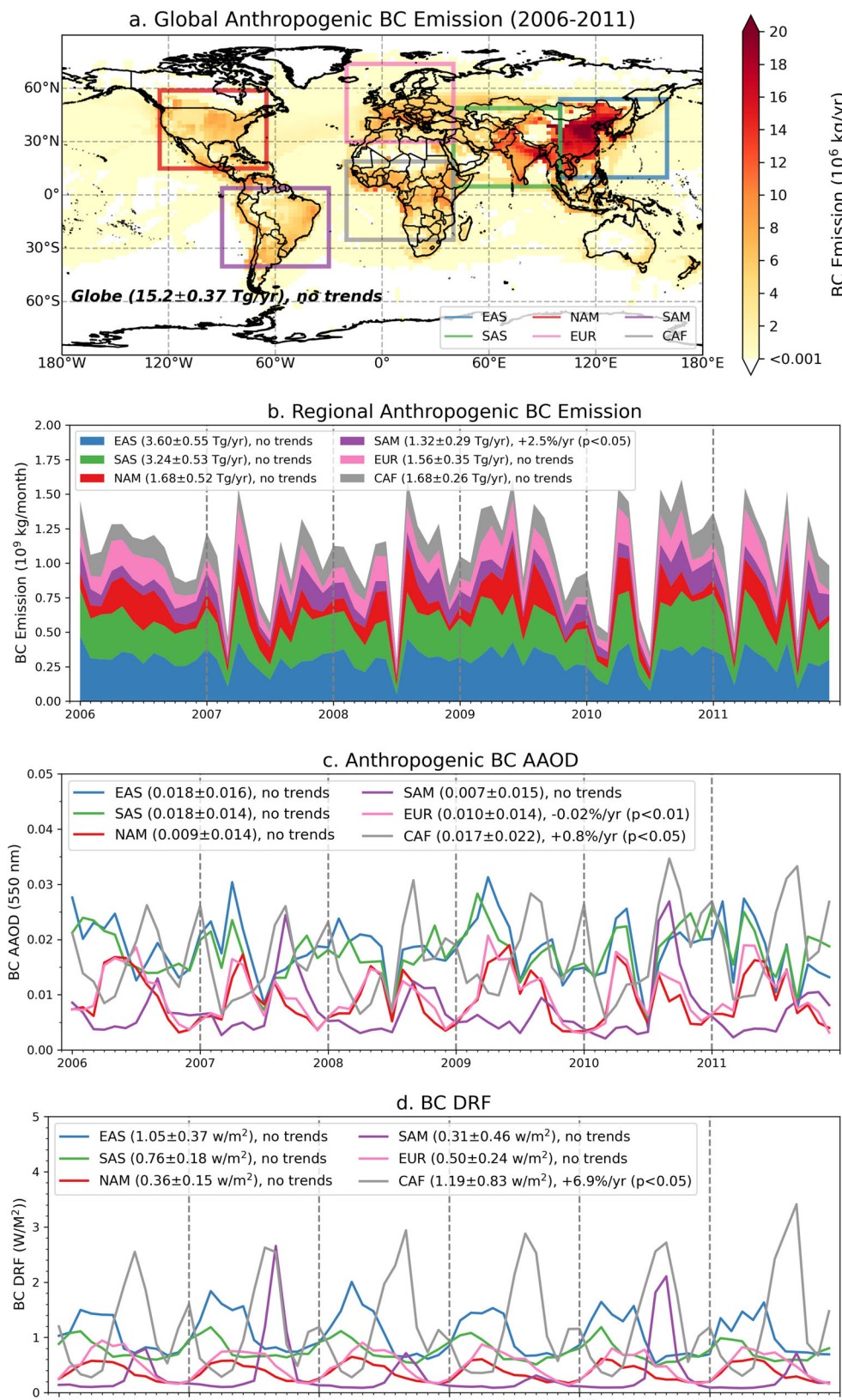

**Fig. 4 | Regional variations in monthly anthropogenic black carbon (BC) emissions and its optical and climate effects from 2006 to 2011 over 6 selected regions: East Asia (ESA), South Asia (SAS), North America (NAM), Europe (EUR), South America (SAM) and Central Africa (CAF). a** Global spatial distribution of annual mean anthropogenic BC emission from 2006 to 2011. **b** Regional variation of monthly anthropogenic BC emission. **c** Monthly variation of regional mean aerosol absorption optical depth (AAOD) at 550 nm due to anthropogenic BC emission. **d** Monthly variation of regional BC direct radiative forcing (DRF) due to anthropogenic BC emission. The regional mean (±1 standard deviation) and the detected linear trends with significance are also present in the upper left.

present over China after the implementation of active clean air policies since 2013[64]. Despite this, we demonstrate the trend analysis based on the MAP-based observationally constrained simulation of aerosol absorption and its radiative effects from 2006–2011. These considerations can be a basis for further trend analysis of aerosol components. Indeed, the current knowledge of aerosol absorption is limited due to scarce global observations in IPCC AR6[8] and several MAP instruments are planned to launch in the near future for enhanced aerosol characterization, e.g. 3MI, CO2M, DPC, PCF, SPEX-one, HARP2 etc[40,65–69].

The monthly variations of anthropogenic BC emissions and its optical and radiative effects from 2006 to 2011 for 6 regions are shown in Fig. 4, and the regional mean ($\pm 1\sigma$) and the detected linear trends with statistical significance (p-value<0.05) are also present. Our best estimate of modern-era anthropogenic BC emission is 15.2 Tg/yr (1 Tg = $10^9$ Kg), $1\sigma$ = 2.35 Tg/yr, and no significant trends detected for the global BC emission. It should be noted that in contrast to the bottom-up emission inventory, the satellite-constrained (top-down) aerosol emission database presents strong interannual variability resulting in less significant annual trends[70]. The regional trend of anthropogenic BC emission (Fig. 4b) is only significant over SAM, that is +2.5% per year. EAS and SAS are the two main regions emitted in total ~6.8 Tg BC per year, which accounts for ~45% global anthropogenic BC emissions. The regional mean anthropogenic BC AAOD (Fig. 4c) show significant decreasing trend over EUR about 0.02% per year, and the increasing trend (+0.8%) is observed significantly over CAF. Additionally, the regional mean anthropogenic BC AAOD over these six industrial regions is much higher (a factor of 2–4) than the global mean (0.0044). A previous study by Zhang et al.[71] found an increasing AAOD trend in the US that is mostly driven by interannual variability of local biomass burning emissions. The regional effects of BC DRF (Fig. 4d) that are caused by anthropogenic BC emission vary from +0.31 W/m² (SAM) to +1.19 W/m² (CAF). The annual increasing trend of BC DRF over CAF is +6.9%, which is significant and noteworthy.

Table 1 summarizes the regional anthropogenic BC AAOD, BC DRF, and BC DRF efficiency based on our simulation from 2006 to 2011. Interestingly, the BC forcing efficiency is relatively small over the industrial regions and varies significantly from region to region (~40–70 W/m² per unit AAOD vs. 105 W/m²). This probably relates to several factors: (i) the fraction of absorbing aerosol below clouds is relatively high over industrial regions; (ii) AOD and AAOD in industrial regions are generally high and DRF is affected stronger by non-linear multiple scattering effects that leads to saturation of the absorption influence and results in lower DRF efficiency[72]. This shows that an assumption of linear increase of DRF with increase AOD and AAOD may lead to significant uncertainties in evaluation of aerosol forcing variability, especially at regional scale; (iii) the six selected regions cover mainly industrial regions and its surrounded vegetation areas where the surface is darker compared to the desert-, snow- or ice-covered areas. Correspondingly, absorbing aerosol above dark surfaces tends to generate smaller direct RF than over bright surfaces. Overall, the relatively low BC forcing efficiency leads to the rather alarming suggestion that the possible reduction of BC emissions in the future over industrial regions may not be as efficient as expected by climate change mitigation. At the same time, the numbers also suggest that the presence of a relatively minor amount BC in remote areas may have a significant DRF effect on climate.

## Discussion

While, the pattern of global aerosol extinction (i.e., AOD) is relatively well represented in the models based on global satellite remote sensing measurements, the amount and distribution of global aerosol absorption (i.e., AAOD and SSA) is quite uncertain due to the lack of sensitivity of the available remote sensing observations to aerosol absorption. In addition, the uncertainties of derived aerosol absorption properties from both satellite- and ground-based remote sensing measurements are highly dependent upon the aerosol loading; for example, the higher aerosol abundance offers more information content and results in lower uncertainty of aerosol absorption inversion[21,22]. Therefore, direct comparisons of GCM simulations to satellite and ground-based AAOD inversion products (that require high aerosol abundance for accuracy) can result in biases and a mis-accounting of global aerosol absorption effects, especially for situations when the aerosol loading is low. The direct use of both AOD and AAOD satellite MAP products for constraining aerosol emissions seems to address the issue, even though we cannot rule out that the uncertainties associated with modeling aerosol processes, meteorology field, and the mass-to-optical conversion that may influence the constraints on emission strength. The corresponding aerosol absorption and its climate effects are well constrained by the MAP products.

The POLDER-3/GRASP generated global satellite AAOD dataset has a confirmed spectral (VIS-NIR) uncertainty of ±0.01, and this provides enhanced information about the global distribution of aerosol absorption that is essential for improving estimates of aerosol climate forcing. We expect to achieve significant refinements in characterizing the global distribution of aerosol absorption and its climate effect by tuning both modeled AOD and AAOD with this MAP-generated spectral data to constrain aerosol emissions. The results obtained with this tuning suggest that modern-era global mid-visible aerosol absorption (AAOD) is at 0.0070 ($1\sigma$ = 0.0002) and BC AAOD at 0.0055 ($1\sigma$ = 0.0003) with high confidence, and is in agreement with two independent assessments using observational constraints from AERONET ground-based measurements. Meanwhile our estimate of global AAOD is higher than the AeroCom Phase II (0.0042, $1\sigma$ = 0.0019) by 67% and Phase III by 30% (0.0054, $1\sigma$ = 0.0023). This implies that the current models tend to underestimate the role that atmospheric aerosol absorption contributes in climate change. Our aerosol DRF results (−0.14 W/m²) further suggest about an 60% increase towards warming with respect to the same model simulation (−0.36 W/m²) that is unconstrainted by aerosol absorption observations. In addition, our results show the modern-era DRF due to the main atmospheric aerosol absorber BC at the level of +0.33 [+0.17, +0.54] W/m;² that is below the IPCC AR5 value of +0.4 [+0.05, +0.8] W/m² and the 95% confidence interval is pinpointed significantly. Furthermore, our estimate of BC DRF is higher than the AeroCom Phase II multi-model mean (+0.23 W/m²), CMIP6 multi-model mean (+0.28 W/m²) and lower than the Bond et al.[9] value of +0.51 W/m² that was obtained by scaling modeled aerosol absorption to ground-based AERONET retrievals. Despite the constrained global BC AAOD being only ~10% difference with Bond et al.[9], the resulted global BC DRF is with a factor of 1.5 discrepancy. This is explained by the spatial variability of BC forcing efficiency which is much smaller (~40–70 W/m² per unit AAOD) over industrial regions than the other areas (105 W/m² per unit AAOD). This hints at the overall importance of the large-scale spatial distribution of aerosol absorption derived from satellite observations, such as multi-angular polarimeters. For the coming MAP era[40], on one hand, we could be of high expectation to constrain and pinpoint global aerosol absorption as well as its regional trends. On the other hand, aerosol climate effects are heavily influenced by its anthropogenic fraction, which can only be obtained through refinement of model simulations supplemented by observational constraints.

## Methods

### Absorbing aerosol emission constrained by POLDER-3/GRASP

The MAP-constrained (MACE) daily absorbing aerosol emission of (BC, OA, and DD) was retrieved by fitting POLDER-3/GRASP spectral AOD and AAOD based on the adjoint GEOS-Chem model[73]. A previous sensitivity study[48] suggested that the uncertainty of the derived emission is 25.8% for DD, 5.9% for BC, and 26.9% for OA, and the mean bias of simulated AAOD compared to observations improved from an

underestimation of 0.012 to -0.000 at 550 nm using the MACE database in the reference year 2010 (Chen et al.[49]). The assumption of the mass to extinction/absorption conversion plays a key role in constraining emission from the satellite-derived aerosol optical properties, especially for the BC mass to an absorption coefficient (MAC) that is of high diversity in the climate models. The MACE database is generated using MAC 6.3 m²/g with refractive index 1.95–0.79i for BC. The microphysical properties for other species are provided in Chen et al.[48]. We emphasize that although the emissions are indeed associated with the mass to extinction/absorption assumptions, the meteorological fields, and the host model treatments of aerosol processes, the simulated global aerosol absorption, as well as its climate effects, are well constrained by observations.

### Separation of anthropogenic and natural emissions

The separation of anthropogenic and natural contributions from the MACE database of total (anthropogenic + natural) emissions is crucial for further climate effects evaluation. However, the direct separation anthropogenic and natural contribution from top–down emission database is generally difficult. In our GEOS-Chem simulation, we keep a priori knowledge of anthropogenic emission ratio of each grid at daily scale and separate MACE daily total BC, OC, and DD emissions into anthropogenic and natural parts. Our GEOS-Chem v11-01 simulation assumes that 20% of DD emissions at each grid box are attributed to anthropogenic origins (Heald et al.[50]). The a priori anthropogenic BC and OC emissions are adopted from Hemispheric Transport of Atmospheric Pollution (HTAP) Phase II inventory[74] and Global Fire Emission Database (GFED) v4s (van der Werf[75].) is used as a priori biomass burning emission database. Even though the anthropogenic emission ratio of each grid is kept the same as apriori at a daily scale, the global anthropogenic emission fraction still change because of the observational constraint on temporal and spatial variability. For example, more BC emission is derived over industrail region where the anthropogenic fraction is high. Supplementary Fig 1 shows the spatial distribution of MACE annual emissions of BC, OC and DD from 2006 to 2011, as well as the monthly variation of anthropogenic and natural emissions of BC, OC, and DD. Our estimation of modern-era global BC emission is 18.4 Tg/yr, and 15.2 Tg/yr of that is emitted from anthropogenic origins. Our estimation of modern-era BC emission is consistent with previous estimate of 14.6–22.2 Tg/yr by Cohen and Wang[76] using AERONET AAOD as observational constraints, even though both of them are at least a factor of 2 higher than current bottom-up inventories. Global OC emission is estimated at 88.7 Tg/yr from anthropogenic sources, and 20.0 Tg/yr from natural sources. The global up to 12 μm (dust particle diameter) DD emission is 731.1 Tg/yr.

### Evaluation of GESO-Chem simulated AAOD with AERONET

We used the daily MACE BC, OC, and DD emissions in the GEOS-Chem (v11-01) model simulation at 2° (latitude) × 2.5° (longitude) resolution, and evaluated the GEOS-Chem simulated daily spectral AAOD against of all available AERONET Version 3 Level 2 (Giles et al.[77]) inversion products of AAOD (Dubovik and King[14]). Supplementary Fig 2 shows the evaluation results at blue (440 nm), mid-visible (550 nm), and near-infrared (870 nm) channels. The gray envelope defines the target region of max 0.01 or 10% AAOD. By evaluating >150,000 matchups globally, the GEOS-Chem model-simulated AAOD show good agreement with AERONET products with >50% pairs satisfying target requirements, and correlation coefficient (R) is -0.52, root-mean-square-error is 0.020 and bias 0.007 at 550 nm. Nevertheless, the bias is still -100% relative to our estimated global mean AAOD (0.0070). These evaluation metrics are close to the state-of-art MAP AAOD products against AERONET (Schutgens et al.[43] and Chen et al.[44]). Note the validation statistics are made based on moderate and high AOD conditions with AERONET AOD (440 nm) higher than 0.4. Essentially, the biases are lower than 0.01 spectrally from blue to NIR channels,

which guarantees the uncertainty of global aerosol absorption estimation at this level that is equivalent to the AERONET direct Sun measurement uncertainty of aerosol extinction (Eck et al.[78]).

### Intercomparison with the simulation using a priori emissions

In order to evaluate the use of MAP-constrained MACE emission on the simulation, we cross-validate the simulated speciated AOD and DRF using a priori emission in 2010 (Heald et al.[50]) and our results using MACE emission based on the same GEOS-Chem RRTMG model in Supplementary Table 1. Note the results with MACE emission in this study are based on an average of 2006 to 2011, and the simulation results with a priori emission (Heald et al.[50]) is performed for year 2010. Specifically, Heald et al.[50] utilize EDGAR v3.2 anthropogenic emission inventory (Olivier et al.[79]) with BC and primary OC emissions adopted from Bond et al.[80]. The dust simulation is based on the DEAD dust entrainment scheme (Zender et al.[81]) couple with the GOCART dust source function from Ginoux et al.[82]. The MACE anthropogenic BC emission (15.2 Tg/yr) is -3.4 times higher than a priori BC emission used in Heald et al.[50], and MACE anthropogenic OC emission is higher than a priori OC emission about a factor of 4.8. Since we keep the same 20% anthropogenic fraction for DD emission, both the total and anthropogenic DD emission from MACE is lower than a priori DD emission used in Heald et al.[50] about a factor of 2.1. Consequently, the simulated anthropogenic BC and OC AOD is higher than Heald et al.[50] about a factor of 8.5, and BC DRF increases by a factor of 4.2 while OC DRF slightly changes from −0.055 W/m² to −0.05 W/m². A factor of 2.1 decrease in DD emission results in an increase of DD DRF a factor of -2 towards warming (−0.053 W/m² to −0.026 W/m²).

## Data availability

The satellite constrained absorbing aerosol emission database and GEOS-Chem model simulation results are deposited at ZENODO (https://doi.org/10.5281/zenodo.6348890). The POLDER/GRASP products are also public available at (https://www.grasp-open.com/products/polder-data-release/).

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

## Acknowledgements

This work is funded by Agence Nationale de la Recherche (ANR - French National Research Agency) under contract ANR-21-PRRD-0001-01 signed by ANR and CNRS (Centre National de la Recherche Scientifique) within the frame of Preservation of R&D employment (France Relance program 2020) and the Laboratory of Excellence CaPPA – Chemical and Physical Properties of the Atmosphere – project under contract ANR-II-LABX-0005-01. D. K. Henze recognizes support from NASA 80NSSC21K1343. C. Chen, Z.Q. Li, and Y. Zhang also recognize support from Open Fund of State Key Laboratory of Remote Sensing Science (Grant number OFSLRSS202008) and National Natural Science Foundation of China (Grant No. 41925019). The authors would like to acknowledge the AERONET, GEOS-Chem, Adjoint GEOS-Chem teams for sharing the data and maintaining the code, and make them available for the community.

## Author contributions

C.C. and O.D. proposed the initial idea for the study. C.C., O.D., T.L., and D.K.H. contributed to the inversion algorithm development. C.C. performed the simulation and conducted the data analysis. All authors discussed the results. C.C. and O.D. wrote the manuscript and G.L.S., M.C., D.K.H., Z.Q.L, Y.D., and Y.Z. provided valuable comments and suggestions.

## Competing interests

The authors declare no competing interests.
