## [Peer Review File · Nature Communications]

Multi-angular polarimetric remote sensing to pinpoint global aerosol absorption and direct radiative forcingReviewer #1 (Remarks to the Author):

The authors present a new estimate of total aerosol shortwave energy absorption in the atmosphere, using constraints from satellite multi-angular polarimeters as input to a global chemical transport and a radiative transfer model. This is a highly relevant study that advances our knowledge of a poorly constrained but crucial quantity in the Earth system, and that should spark some discussions given the relatively high values they report compared to contemporary model results. I recommend publication, but I do have a number of questions and comments that the authors may want to address first.

Major comments:

* A first, hopefully simple, comment is that the presentation would be stronger if it was updated to the most recent studies. The authors refer to AR5 throughout, and to AeroCom Phase II, but there has now been both an AR6 (albeit with less detail on BC, but still with very relevant conclusions in Chapter 6; Szopa et al.), and an AeroCom Phase III. Here AAOD has been studied both as part of an overall evaluation (Gliss et al. 2021), and as its own experiment (Sand et al. 2021). I would expect these references to form the starting point of the discussion, at least for comparison with modelling, rather than the older studies.

* Relatedly, I note that on line 45, the authors still state that BC is the second largest contributor to global warming, as claimed by Bond et al. 2013. This is not the case, as has been shown in a number of studies, including the IPCC AR6 (see already Figure 2 of the SPM, or Chapter 7, Forster et al.). BC has had a warming contribution over the historical era of around 0.1 deg C. The reason for the apparent discrepancy here is the difference between RF and ERF, where the ERF of BC - which is more directly related to surface temperature change - is markedly reduced because of rapid adjustments. However these rapid adjustments are there precisely because of the shortwave absorption, and as such are a key reason for why the present study is important. Therefore I urge the authors to add some discussion about RF vs ERF, and rapid adjustments, to the manuscript, to help the reader. It is currently not mentioned at all, as far as I can tell, and all forcing results are in (direct) RF.

* The authors use a specific constrained emission inventory as input to GEOS-Chem, and from this derive the radiative impacts of absorbing aerosols. They also compare (in Methods) to a simulation using standard emissions. However, there is no discussion of the sensitivity of the results to the optical parameters assumed in the various steps of the logic. For instance the BC MAC of 6.3 m²/g seems low compared to recent studies, in particular when considering that most of the BC considered here would be aged, which enhances the absorption. (See e.g. Zanatta or Samset 2018.) In order to compare the results to e.g. the AeroCom spread, which is driven in part by model differences in both optical parameters (for all three absorbing species) and transport, ageing and removal processes, it would be good to at least discuss the overall performance of GEOS-Chem compared to these other models.

* For the trend analysis on line 262 and onwards, I wonder if the relatively short five year period warrants this kind of presentation. Especially given the well documented changes in Asian emissions for the period just after the five years studied here, which would change the picture radically. The composition and monthly diversity are interesting and relevant, but I'm not quite sure what conclusions to draw from the trends documented here?

Minor comments:

Abstract: The first line (18-20) seems to say that solar absorption is uncertain because we don't have enough information about solar absorption. Rephrase? I think the point is that we don't know where the absorbing aerosols are, and in what amounts, and hence we can also not quantify their total shortwave absorption?

I40: "aerosols cool" -> "aerosol have a net cooling effect"? In general, they do both...

I61: "...too weak." Relative to what? Since your premise is that we don't know what the absorption is, you need a qualifier (relative to AERONET?)

I99: POLDER is not explained until Methods. Put in a brief explanation?

I116: "...we obtain the..." This isn't explained until Methods, so maybe a quick recap here and a reference to Methods?

I135: "present-day" 2006-2011 is over a decade ago now, with major changes ongoing in aerosol emissions, so maybe "modern era" or similar?

I212: "...a factor of 4." Where does this increase come from? Is it only the emission inventory, in amount and location? Is the vertical profile affected? The main emission locations? (I know it's touched on in Methods, but maybe recap here?)

Figure 2: Consider including the constrained estimate from Wang et al. 2016 for BC DRF?

References:

Gliss et al. 2021: Atmos. Chem. Phys., 21, 87–128, 2021 <https://doi.org/10.5194/acp-21-87-2021>

Sand et al. 2021: Atmos. Chem. Phys., 21, 15929–15947, 2021 <https://doi.org/10.5194/acp-21-15929-2021>

Zanatta et al. 2016: Europe. Atmos Environ. 2016;145:346–64. <https://doi.org/10.1016/j.atmosenv.2016.09.035>.

Samset et al. 2018: Current Climate Change Reports, <https://doi.org/10.1007/s40641-018-0091-4>

Wang et al. 2016: Journal of Geophysical Research: Atmospheres, 2016, 121 (10), pp.5948-5971. [10.1002/2015JD024326](https://doi.org/10.1002/2015JD024326)

Reviewer #2 (Remarks to the Author):

This paper reports (A) absorbing aerosol optical depth (AAOD) retrieved from spaceborne platform (POLDER); (B) use of that AAOD plus an atmospheric transport model to constrain concentration fields and estimate emission; (C) use of the concentrations in the atmospheric transport model to infer direct radiative forcing (DRF) of black carbon and of anthropogenic aerosol.

Because emission rates are not well known, model-simulated fields could be incorrect, and the inferred climate forcing also incorrect, so it is important to check those modeled concentration fields against observations. However, available observations have been ground-based and therefore measured only at points. The authors' work is the first to use POLDER fields with global coverage to constrain emissions of black carbon and other absorbing aerosols. They have previously published a paper (their ref 45 in 2019) with the emission totals and also the AAOD values (steps A and B above). Of note, authors' prior work has agreed with other studies that showed there is more AAOD in the real atmosphere than is simulated in models, especially attributable to black carbon. That is, estimated effect of black carbon (and other absorption) is higher than models would predict, if only the simulation was used.

 This paper adds the significant step of estimating radiative forcing caused by absorbing aerosol: step (C) above. As this step has not appeared in the literature, I think this paper deserves publication.

 However, some attention is needed to the analysis before I would consider it publishable. My major critiques:

1. The error bars presented for DRF are far too narrow. It implies overconfidence which is not supported by the analysis.
2. The major contribution of this work (the leap from constrained AAOD fields to radiative forcing by BC/OM/Dust) is inconsistent with previous work. It implies a different relationship between AAOD and forcing. I am not questioning the validity of this result, but since this is the main new contribution, it should be explored and supported.
3. Along similar lines as #2, the presentation also sneaks in an estimate of total aerosol radiative forcing including non-absorbing aerosol, but the major factors in constraining these other aerosol types are not discussed.

#1 Error Bars:

The confidence intervals given for anthropogenic DRF are very tight. For black carbon (BC) the value is about 10% of the mean. This is the same uncertainty given for the AAOD, which means there is no uncertainty in forcing caused by any other factor. This implies that:

- a. The anthropogenic fraction is precisely known; in fact authors used their own a priori emissions to apportion the anthropogenic fraction, but if the emissions begin incorrect, this would introduce an uncertainty. This alone could be greater than 10%. The anthropogenic fraction gets even more uncertain when considering desert dust.
- b. the collocation between aerosols and clouds, which affects DRF, is precisely known, but this isn't discussed and I doubt it is good to 10%. As POLDER products have been used for the exact purpose of determining where aerosols lie relative to clouds, more could be said about this.
- c. the optical properties that affect radiative transfer are precisely known; in fact for non-absorbing aerosol the size distribution and backscatter fraction are important. It could be that POLDER constrained some of these assumptions, but it is not stated. (I don't think 'pinpoint' is a good word for the title.)

#2 AAOD to DRF:

This is discussed in Lines 218-227. I will review issues for BC only but they are true of the other absorbing species as well.

In model studies AAOD is found proportional to BC concentration and DRF is also proportional to BC concentration, thus AAOD and DRF may be assumed proportional. However this study estimates that AAOD is higher than ref 6 yet DRF is much lower, and conversely AAOD is much higher (67%) than AEROCOM but forcing is similar. Therefore the AAOD to DRF ratio is different in this study than in previous studies.

Authors acknowledge this issue in these lines. Yet although this is the main novel finding of this manuscript (in my view), the only discussion is this: "the satellite products provide additional details in aerosol spatial distribution that are non-negligible for radiative effect estimates and hardly seen from the ground-based network." If this is the case, then it is important and needs to be isolated and also given as a challenge to future modeling efforts! Has POLDER placed more absorbing aerosol below clouds where it has less forcing? Or in different regions or over ocean where there is less bright surface? A reduction of 50-70% in forcing efficiency that is constrained by spaceborne measurement would be quite notable, but it is not emphasized. Authors have all the data to investigate this question – model fields and observed fields.

This paper also mentions absorption cross section. A cross section is assumed that ignores internal mixing of particles which increases absorption. I assume that since the

model is adjusted to match the AAOD, this choice doesn't matter: if the correct amount of absorption is in the atmosphere, the forcing should be approximately correct, even if the mass is wrong and the absorption per mass is wrong. This however is not stated.

#3 total radiative forcing:

This step requires a division of the total aerosol optical depth among other species. This is far more difficult than dividing among absorbing aerosol components (BC/OM/dust) which is not easy in itself. This division then propagates forward to the assumed optical properties of each species. As stated under #1, the optical properties (especially backscatter fraction) are key in determining negative forcing. I think this part should either be deleted, or it needs a far more expanded discussion of assumptions and uncertainties.

Other comments

I suggest also comparison with Cohen and Wang, <https://doi.org/10.1002/2013JD019912>, which also used ground-based constraints to estimate BC emission.

I am puzzled by the title which describes only the portion that has already been published, i.e. authors' paper in 2019 covered emission and aerosol absorption.

Editorial comment: The paper is generally well written. However I find distracting, the practice of (for example) referring to ref 26. It doesn't read well, isn't common practice and also doesn't allow the reader to identify which body of work compares well or poorly with the current paper, as readers generally do not remember which methods are associated with each reference. Authors seem willing to provide acronyms and identification for their own sequence of work and I suggest they do so for others.

Reviewer #1:

The authors present a new estimate of total aerosol shortwave energy absorption in the atmosphere, using constraints from satellite multi-angular polarimeters as input to a global chemical transport and a radiative transfer model. This is a highly relevant study that advances our knowledge of a poorly constrained but crucial quantity in the Earth system, and that should spark some discussions given the relatively high values they report compared to contemporary model results. I recommend publication, but I do have a number of questions and comments that the authors may want to address first.

Response:

Thank you very much for the positive and constructive comments on our manuscript. Please find our point-by-point response below.

Major comments:

* A first, hopefully simple, comment is that the presentation would be stronger if it was updated to the most recent studies. The authors refer to AR5 throughout, and to AeroCom Phase II, but there has now been both an AR6 (albeit with less detail on BC, but still with very relevant conclusions in Chapter 6; Szopa et al.), and an AeroCom Phase III. Here AAOD has been studied both as part of an overall evaluation (Gliss et al. 2021), and as its own experiment (Sand et al. 2021). I would expect these references to form the starting point of the discussion, at least for comparison with modelling, rather than the older studies.

Response:

Thanks very much for the constructive suggestion! The manuscript was prepared before the release of the latest reports (e.g. AR6 and AeroCom Phase III). We fully agree that the adoption of the latest evaluation will add value to our work. In the revised manuscript, we have included the latest results from AR6 (Szopa et al., 2021; Forster et al., 2021), AeroCom Phase III (Gliß et al., 2021; Sand et al., 2021), as well as some other key relevant studies, such as Kinne 2019 (MACv2), Wang et al. 2016, and Thornhill et al. 2021 (CMIP6 models). Meanwhile, we also kept and intercompared with the results from AR5 and AeroCom Phase II to show the evolution of aerosol climate effects evaluation.

Szopa, S. *et al.* Short-lived Climate Forcers. in *Climate Change 2021: The Physical Science Basis. Contribution of Working Group I to the Sixth Assessment Report of the*

Intergovernmental Panel on Climate Change (eds. Masson-Delmotte, V. et al.) 817–922 (Cambridge University Press, 2021). doi:10.1017/9781009157896.008.

Forster, P. *et al.* The Earth's Energy Budget, Climate Feedbacks and Climate Sensitivity. in *Climate Change 2021: The Physical Science Basis. Contribution of Working Group I to the Sixth Assessment Report of the Intergovernmental Panel on Climate Change* (eds. Masson-Delmotte, V. et al.) 923–1054 (Cambridge University Press, 2021).

doi:10.1017/9781009157896.009.

Gliß, J. *et al.* AeroCom phase III multi-model evaluation of the aerosol life cycle and optical properties using ground- And space-based remote sensing as well as surface in situ observations. *Atmospheric Chemistry and Physics* **21**, 87–128 (2021).

Sand, M. *et al.* Aerosol absorption in global models from AeroCom Phase III. *Atmospheric Chemistry and Physics* **21**, 15929–15947 (2021).

Thornhill, G. D. *et al.* Effective radiative forcing from emissions of reactive gases and aerosols-A multi-model comparison. *Atmospheric Chemistry and Physics* **21**, 853–874 (2021).

Kinne, S. Aerosol radiative effects with MACv2. *Atmospheric Chemistry and Physics* **19**, 10919–10959 (2019).

Wang, R. *et al.* Estimation of global black carbon direct radiative forcing and its uncertainty constrained by observations. *Journal of Geophysical Research: Atmospheres* **121**, 5948–5971 (2016).

* Relatedly, I note that on line 45, the authors still state that BC is the second largest contributor to global warming, as claimed by Bond et al. 2013. This is not the case, as has been shown in a number of studies, including the IPCC AR6 (see already Figure 2 of the SPM, or Chapter 7, Forster et al.). BC has had a warming contribution over the historical era of around 0.1 deg C. The reason for the apparent discrepancy here is the difference between RF and ERF, where the ERF of BC - which is more directly related to surface temperature change - is markedly reduced because of rapid adjustments. However these rapid adjustments are there precisely because of the shortwave absorption, and as such are a key reason for why the present study is important. Therefore I urge the authors to add some discussion about RF vs ERF, and rapid adjustments, to the manuscript, to help the reader. It is currently not mentioned at all, as far as I can tell, and all forcing results are in (direct) RF.

Response:

Indeed, in this study, we were focused on the instantaneous direct aerosol radiative forcing (IRFari). The effective radiative forcing (ERF) by taking into account rapid adjustment effects is not our main goal here. Even though, many recent studies showed that BC ERF is about half of its corresponding IRF due to largely the surface temperature adjustment (Smith et al., 2018). As a consequence, the warming effects (positive forcing) by BC is significantly reduced ~50%, which is no longer as crucial as it was. However, the derivation of BC or generally aerosol induced forcing adjustment strongly depends on the estimation of its shortwave absorption and especially on its precise global spatial distribution. However, the current rapid adjustment is derived using spatially equal scaling assuming accurate variability of aerosol, that is generally high uncertain. Also, the knowledge of those adjustment specifically for aerosol absorption is still rather limited. Therefore, there is clear ground for questioning the accuracy the used adjustment and correspondingly the accuracy of estimated effective forcing.

In any case, we find these specific comment and suggestion very interesting! We've changed the relevant statement and added some descriptions in the revised manuscript as following:

“Although the aerosol absorption by black carbon (BC) aerosol is known to be one of the largest contributors with carbon dioxide (CO₂) and methane (CH₄) for heating our planet^{4,5,6}, there are still significant challenges for pinpointing the effects of absorbing aerosols⁷. The recent AR6⁸ reports about a 50% reduction of BC warming effects by adjusting its rapid climate responses. Nonetheless the rapid adjustments strongly rely on the highly uncertain simulation of global aerosol absorption magnitude, especially its spatial distribution.”.

Smith, C. J. *et al.* Understanding Rapid Adjustments to Diverse Forcing Agents. *Geophysical Research Letters* **45**, 12,023-12,031 (2018).

* The authors use a specific constrained emission inventory as input to GEOS-Chem, and from this derive the radiative impacts of absorbing aerosols. They also compare (in Methods) to a simulation using standard emissions. However, there is no discussion of the sensitivity of the results to the optical parameters assumed in the various steps of the logic. For instance the BC MAC of 6.3 m²/g seems low compared to recent studies, in particular when considering that most of the BC considered here would be aged, which enhances the absorption. (See e.g. Zanatta or Samset 2018.) In order to compare the results to e.g. the AeroCom spread, which is driven in part by model differences in both optical paramters (for all three absorbing

species) and transport, ageing and removal processes, it would be good to at least discuss the overall performance of GEOS-Chem compared to these other models.

Response:

Thanks for pointing out this issue. Indeed, we presented a single model study, where the host model treatment of transport, ageing, removal processes, meteorology etc. will play a role in the assessment that multi-model mean/median could dilute this effect to some extent. We refer to the study by Heald et al. (2014) where the comprehensive analysis of the GEOS-Chem standard simulation with AeroCom Phase II and Phase I means are present. Basically, on one hand the main characteristics (burden, MEE, TOA DRF etc.) agrees generally within ~ 1 sigma interval. On the other hand, some different assumptions were used, for example, GEOS-Chem assumes 20% anthropogenic desert dust emissions that is different in other models. Our GEOS-Chem simulation is based on the similar assumptions as by Heald et al. (2014) with updated BC, OC and DD emissions. We fully agree that the BC MAC $6.3 \text{ m}^2/\text{g}$ adopted in our scheme is lower than in many studies. At the same time, it is within the 1 sigma lower limit of the recommendation $7.5 \pm 1.2 \text{ m}^2/\text{g}$ by Bond and Bergstrom (2006) for fresh BC. In addition, the used assumption of externally mixture ignores the BC absorption enhancement and results in the overestimation of our BC emissions. Nonetheless, since our emission database is adjusted to match the spectral AAOD from POLDER-3/GRASP observations, the amount of absorption in the atmosphere then is well constrained even though the emission/mass is highly uncertain depending on the MAC as well as absorption enhancement. Thus, we consider that the global aerosol absorption and the radiative effects due to aerosol absorption are well constrained and approximately in spite of used assumptions.

Bond, T. C. & Bergstrom, R. W. Light Absorption by Carbonaceous Particles: An Investigative Review. *Aerosol Science and Technology* **40**, 27–67 (2006).

Heald, C. L. *et al.* Contrasting the direct radiative effect and direct radiative forcing of aerosols. *Atmos. Chem. Phys* **14**, 5513–5527 (2014).

* For the trend analysis on line 262 and onwards, I wonder if the relatively short five year period warrants this kind of presentation. Especially given the well documented changes in Asian emissions for the period just after the five years studied here, which would change the picture radically. The composition and monthly diversity are interesting and relevant, but I'm not quite sure what conclusions to draw from the trends documented here?

Response:

We fully agree that the trend analysis is very limited and it is difficult to draw interesting results from only six years (2006 – 2011) of data that is also a bit far from the present. In addition, the emission has significantly changed after 2010 over East Asia (Samset et al., 2019). Specifically, China has implemented active clean air policies from 2013, and the aerosol bottom-up emissions show significant decline trend since then (Zheng et al., 2018). Nonetheless, as discussed in IPCC AR6 (Szopa et al., 2021), the observational based analysis of aerosol and aerosol components trends is challenging and uncertain due to scarce global observations. From this point of view, the section demonstrated a good basis for the possible analysis of aerosol absorption and radiative effects trends from MAP (multi-angular polarimeters) observations that is of high interest considering the era of MAP instruments coming in near future for aerosol characterization, e.g. 3MI, CO2M, DPC, PCF, SPEXone, etc. (Dubovik et al., 2019; Fougnie et al., 2018; Li et al., 2018, 2022; Hasekamp et al., 2019; Remer 2019). We've adjusted our text to focus on the future possibility instead of interpreting trends from 2006 to 2011.

Dubovik, O. *et al.* Polarimetric remote sensing of atmospheric aerosols: Instruments, methodologies, results, and perspectives. *Journal of Quantitative Spectroscopy and Radiative Transfer* **224**, 474–511 (2019).

Fougnie, B. *et al.* The multi-viewing multi-channel multi-polarisation imager – Overview of the 3MI polarimetric mission for aerosol and cloud characterization. *Journal of Quantitative Spectroscopy and Radiative Transfer* **219**, 23–32 (2018).

Hasekamp, O. P. *et al.* Aerosol measurements by SPEXone on the NASA PACE mission: expected retrieval capabilities. *Journal of Quantitative Spectroscopy and Radiative Transfer* **227**, 170–184 (2019).

Li, Z. *et al.* Directional Polarimetric Camera (DPC): Monitoring aerosol spectral optical properties over land from satellite observation. *Journal of Quantitative Spectroscopy and Radiative Transfer* **218**, 21–37 (2018).

Li, Z. *et al.* The polarization crossfire (PCF) sensor suite focusing on satellite remote sensing of fine particulate matter PM_{2.5} from space. *Journal of Quantitative Spectroscopy and Radiative Transfer* **286**, 108217 (2022).

Remer, L. A. *et al.* Retrieving Aerosol Characteristics From the PACE Mission, Part 2: Multi-Angle and Polarimetry. *Frontiers in Environmental Science* **7**, 94 (2019).

Samset, B. H., Lund, M. T., Bollasina, M., Myhre, G. & Wilcox, L. Emerging Asian aerosol patterns. *Nature Geoscience* **12**, 582–584 (2019).

Szopa, S. *et al.* Short-lived Climate Forcers. in *Climate Change 2021: The Physical Science Basis. Contribution of Working Group I to the Sixth Assessment Report of the Intergovernmental Panel on Climate Change* (eds. Masson-Delmotte, V. *et al.*) 817–922 (Cambridge University Press, 2021). doi:10.1017/9781009157896.008.

Zheng, B. *et al.* Trends in China’s anthropogenic emissions since 2010 as the consequence of clean air actions. *Atmospheric Chemistry and Physics* **18**, 14095–14111 (2018).

Minor comments:

Abstract: The first line (18-20) seems to say that solar absorption is uncertain because we don't have enough information about solar absorption. Rephrase? I think the point is that we don't know where the absorbing aerosols are, and in what amounts, and hence we can also not quantify their total shortwave absorption?

Response:

Thanks! We’ve rephrased the first sentence to “Quantitative estimations of shortwave aerosol absorption are rather uncertain due to the lack of reliable information about the global distribution of atmospheric aerosols.”.

140: "aerosols cool" -> "aerosol have a net cooling effect"? In general, they do both...

Response:

Corrected.

161: "...too weak." Relative to what? Since your premise is that we don't know what the absorption is, you need a qualifier (relative to AERONET?)

Response:

We’ve revised the sentence to “Generally, most climate models simulate aerosol absorption that is notably weaker compared to the values derived (inverted) from remote sensing measurements such as AERONET.”.

199: POLDER is not explained until Methods. Put in a brief explanation?

Response:

We've added a short description of POLDER instrument in the revised manuscript line 107: *"POLDER (Polarization and Directionality of the Earth's Reflectances) was developed to measure spectral directional polarized solar radiance reflected by the Earth-atmosphere system (Deschamps et al., 1994). POLDER-3 instrument on board PARASOL satellite was the longest to date operational space-borne MAP sensor, while POLDER-1 and 2 have rather limited time series of measurements^{38,40}."*

1116: "...we obtain the..." This isn't explained until Methods, so maybe a quick recap here and a reference to Methods?

Response:

Thanks! We revised with a short illustration:

"We follow the similar GEOS-Chem scheme (meteorology, physical and chemical processes) as Heald et al. (2014) where an early version v9-01 was used. Moreover, by replacing the a priori emission inventories with MAP-constrained 6 years (2006-2011) BC, OA and DD daily emissions from MACE, we obtain the observationally constrained aerosol absorption simulation (see the Method section)."

1135: "present-day" 2006-2011 is over a decade ago now, with major changes ongoing in aerosol emissions, so maybe "modern era" or similar?

Response:

Thanks for the suggestion! We also feel a bit inappropriate to use "present-day" for year 2006-2011, although the latest AEROCOM Phase III uses the year 2010 as a reference for the "present-day" assessment, and CMIP6 multi-model assessment is based on 1850 to 2014. In the revised manuscript we used term "modern-era" emphasizes the used period of 2006-2011 years.

1212: "...a factor of 4." Where does this increase come from? Is it only the emission inventory, in amount and location? Is the vertical profile affected? The main emission locations? (I know it's touched on in Methods, but maybe recap here?)

Response:

Thanks for the suggestion!

Figure 2: Consider including the constrained estimate from Wang et al. 2016 for BC DRF?

Response:

Thanks for the information! We've added Wang et al. (2016) estimation in Figure 2 and our intercomparison discussion.

References:

- Gliss et al. 2021: Atmos. Chem. Phys., 21, 87–128, 2021 <https://doi.org/10.5194/acp-21-87-2021>
- Sand et al. 2021: Atmos. Chem. Phys., 21, 15929–15947, 2021 <https://doi.org/10.5194/acp-21-15929-2021>
- Zanatta et al. 2016: Europe. Atmos Environ. 2016;145:346–64. <https://doi.org/10.1016/j.atmosenv.2016.09.035>.
- Samset et al. 2018: Current Climate Change Reports, <https://doi.org/10.1007/s40641-018-0091-4>
- Wang et al. 2016: Journal of Geophysical Research: Atmospheres, 2016, 121 (10), pp.5948-5971. 10.1002/2015JD024326

Reviewer #2:

This paper reports (A) absorbing aerosol optical depth (AAOD) retrieved from spaceborne platform (POLDER); (B) use of that AAOD plus an atmospheric transport model to constrain concentration fields and estimate emission; (C) use of the concentrations in the atmospheric transport model to infer direct radiative forcing (DRF) of black carbon and of anthropogenic aerosol.

Because emission rates are not well known, model-simulated fields could be incorrect, and the inferred climate forcing also incorrect, so it is important to check those modeled concentration fields against observations. However, available observations have been ground-based and therefore measured only at points. The authors' work is the first to use POLDER fields with global coverage to constrain emissions of black carbon and other absorbing aerosols. They have previously published a paper (their ref 45 in 2019) with the emission totals and also the AAOD values (steps A and B above). Of note, authors' prior work has agreed with other studies that showed there is more AAOD in the real atmosphere than is simulated in models, especially attributable to black carbon. That is, estimated effect of black carbon (and other absorption) is higher than models would predict, if only the simulation was

used.

 This paper adds the significant step of estimating radiative forcing caused by absorbing aerosol: step (C) above. As this step has not appeared in the literature, I think this paper deserves publication.

Response:

The authors appreciate the comprehensive summary, positive comments and helpful advices on our manuscript. Please find our point-by-point response below.

 However, some attention is needed to the analysis before I would consider it publishable.

My major critiques:

1. The error bars presented for DRF are far too narrow. It implies overconfidence which is not supported by the analysis.

Response:

Thanks very much for pointing out this issue! Our error bars are represented by the standard deviation of 6 years (2006-2011) global annual mean. Indeed, the yearly deviation of AAOD, emissions and DRF are very small, which makes our confidence interval unrealistically narrow. We fully agree that our uncertainty estimates are overconfident and certainly underestimate the uncertainty of these fields in reality, and they are difficult to compare directly with the other studies adopted from multi-model mean/median, sensitivity analysis and expert decisions. On the other hand, our study is a single model assessment. The derivation of confidence interval from multi-model distributions, which could dilute some biases from individual model, is impossible. Nonetheless, we have tried to improve our estimates in the revised manuscript by adjusting our strategy for obtaining the confidence interval. First of all, the global aerosol absorption is constrained by the POLDER-3/GRASP spectral AAOD products which is of relative high confidence. Therefore, we kept our estimation of global aerosol absorption 0.0070 from annual mean, with a 95% confidence interval [0.0068, 0.0073]. Second, the anthropogenic fractions, obtained using a priori information from bottom-up emission inventories are of medium/low confidence. We used monthly variations for estimating the uncertainty in values used for the anthropogenic fractions. For example, the BC anthropogenic fraction now can deviate from 80.3% in the range of 65.9% to 88.9% in comparison with the range of 77.6% to 83.2% suggested previously. Based on the modifications, we estimated the following means and confidence

intervals for anthropogenic BC: AAOD 0.0044 [0.0025, 0.0062], BC DRF +0.33 W/m² [+0.17, +0.54] and aerosol DRF -0.14 W/m² [-0.25, +0.013].

2. The major contribution of this work (the leap from constrained AAOD fields to radiative forcing by BC/OM/Dust) is inconsistent with previous work. It implies a different relationship between AAOD and forcing. I am not questioning the validity of this result, but since this is the main new contribution, it should be explored and supported.

Response:

We checked the forcing efficiency from our simulation in this stage, and intercompared with previous studies. Indeed, we identified some interesting aspects. Basically, our global mean BC forcing efficiency (~ 75 W/m²) is lower than in other studies by about $\sim 30\%$ to $\sim 100\%$ (AeroCom Phase II 133 W/m² [84.3, 216]; MACv2 ~ 150 W/m²; Bond2013 104 W/m²). It implies that we obtained more emissions and stronger absorption over industrial regions, where the forcing efficiency is lower than the rest of globe (see in Answers #2). At the same time, the radiative flux calculations showed that common assumption of linear increase of DRF with increase AOD and AAOD may lead to significant uncertainties in evaluation of aerosol forcing. For example, the current studies suggest that BC RF efficiency (shown in Figure R2) varies over polluted regions and is notably different from rest of world. Therefore, using a single number of BC RF efficiency is certainly leads to uncertainties in RF estimation. Moreover, the efficiency values from Figure R2 reveals that BC RF efficiency over industrial regions is nearly twice lower than over other regions. This probably relates to the fact that AOD and AAOD in industrial areas are generally high and RF is affected stronger by non-linear multiple scattering effects resulting in lower RF efficiency (e.g., see Derimian et al., 2016). It should be noted that this observation potentially leads to rather alarming suggestion that the possible reduction of BC emissions in the future over industrial regions may not be as efficient as expected by climate change mitigation. At the same time, the numbers also suggest that presence relatively minor amount of BC in remote areas may have a significant RF effect on climate.

Thanks for the constructive suggestion! We've added this discussion in the revised manuscript.

Derimian, Y. *et al.* Comprehensive tool for calculation of radiative fluxes: illustration of shortwave aerosol radiative effect sensitivities to the details in aerosol and underlying surface characteristics. *Atmospheric Chemistry and Physics* **16**, 5763–5780 (2016).

3. Along similar lines as #2, the presentation also sneaks in an estimate of total aerosol radiative forcing including non-absorbing aerosol, but the major factors in constraining these other aerosol types are not discussed.

Response:

Indeed, in our scheme using POLDER/GRASP products to constrain emissions of aerosol species, we inverted spectral AAOD (uniquely available from POLDER/GRASP products) to retrieve 3 absorbing aerosol species while we assumed the 2 non-absorbing components (SS: sea salt and SNA: sulfate-ammonium-nitrate) remained the same as in a prior GEOS-Chem simulation. This approach also helped to reduce the number of unknowns in the state vector and therefore helped to stabilize the retrieval solution. However, it is clear that the SNA emissions are probably highly uncertain over industrial regions that may affect the retrieval of OC emission significantly. In order to evaluate this effect, we concluded a sensitivity analysis that showed that 100% underestimation of SNA emission led to a factor of 1.5 overestimation of OC, while the effects on BC and DD remain largely within 10% from true values. In addition to avoid strong misrepresentation of OC by fitting total AOD with fixed SNA, we use two-step approach. First, we fitted only PARASOL/GRASP spectral AAOD to retrieve BC, OC and DD emissions. Then, as a second step, we adjusted the retrieved emission by fitting spectral total AOD corresponding to Ångström exponent (AE) smaller than 1.0. The second step allowed more adequate accounting for the coarse mode dominant aerosol that manifest themselves less strongly in AAOD. This two-step approach helps to reduce potential errors in OC emission retrieval. For example, synthetic test showed that using two-step approach even in the extreme condition with 100% underestimation of SNA and SS, the uncertainties of retrieved monthly total BC, OC and DD emissions remains within 10% for DD, 5% for BC and 12% for OC (Chen et al., 2019). Therefore, in present study, we kept a prior GEOS-Chem simulation of SNA (-0.39 W/m^2) and SS (0 W/m^2) for determination of total aerosol DRF, which are in line with AEROCOM Phase II results (SNA: -0.40 W/m^2 ; SS: 0 W/m^2) (Myhre et al., 2013). In this respect, the general assumption of our study is that the major changes in total aerosol DRF are due to the updates of MAP-constrained absorbing species. We've added this discussion in the text.

Chen, C. *et al.* Constraining global aerosol emissions using POLDER/PARASOL satellite remote sensing observations. *Atmospheric Chemistry and Physics* **19**, 14585–14606 (2019).

Myhre, G. *et al.* Radiative forcing of the direct aerosol effect from AeroCom Phase II simulations. *Atmos. Chem. Phys* **13**, 1853–1877.

#1 Error Bars:

The confidence intervals given for anthropogenic DRF are very tight. For black carbon (BC) the value is about 10% of the mean. This is the same uncertainty given for the AAOD, which means there is no uncertainty in forcing caused by any other factor. This implies that:

- a. The anthropogenic fraction is precisely known; in fact authors used their own a priori emissions to apportion the anthropogenic fraction, but if the emissions begin incorrect, this would introduce an uncertainty. This alone could be greater than 10%. The anthropogenic fraction gets even more uncertain when considering desert dust.
- b. the collocation between aerosols and clouds, which affects DRF, is precisely known, but this isn't discussed and I doubt it is good to 10%. As POLDER products have been used for the exact purpose of determining where aerosols lie relative to clouds, more could be said about this.
- c. the optical properties that affect radiative transfer are precisely known; in fact for non-absorbing aerosol the size distribution and backscatter fraction are important. It could be that POLDER constrained some of these assumptions, but it is not stated.

(I don't think 'pinpoint' is a good word for the title.)

Response:

Thanks very much for pointing out the issue and constructive suggestions!

It's rather difficult to estimate the uncertainty from our single model assessment. We've reviewed and modified our strategy (see in the 1st response). Basically, we allowed larger deviation of anthropogenic fraction which is mainly obtained from our a priori emission inventories of low confidence. The BC anthropogenic fraction now can deviate from 80.3% in the range of 65.9% to 88.9% in comparison with the range of 77.6% to 83.2% suggested previously. The estimate certainly provides better representation for BC anthropogenic fraction uncertainty (Figure R1). For example, Kinne *et al.* (2019) used 50%, 70% and 85% anthropogenic BC fractions to further estimate BC and aerosol DRF. On the other hand, our estimation of modern-era aerosol absorption is constrained by the observations from POLDER-3/GRASP that is of high confidence. Therefore, we kept the estimation of modern-era AAOD 0.0070 with the confidence interval of [0.0068, 0.0073] that seems consistent with

MACv2 0.0072 value that is obtained using AERONET AAOD. Generally, we revised all estimations associated with anthropogenic activities by considering larger anthropogenic fraction confidence interval than previous estimate. As a consequence, our revised estimations of means and confidence intervals are 0.0051 [0.0030, 0.0070] for anthropogenic AAOD, for BC AAOD 0.0044 [0.0025, 0.0062], for BC DRF +0.33 W/m² [+0.17, +0.54] and for aerosol DRF -0.14 W/m² [-0.25, +0.013].

Regarding to the title, we have decided to refine it and reduce to “Multi-angular polarimetric remote sensing to pinpoint global aerosol absorption”. We would like to keep the word “pinpoint”, since we refer to global aerosol absorption. In this respect, even though the resulted anthropogenic climate effects are still of high uncertain due to the anthropogenic fraction is not precisely known, the AAOD overall is rather strongly constrained by the MAP aerosol products.

Figure R1. Estimation of global anthropogenic fraction of AOD, AAOD and BC AAOD. The boxplots represent all mean estimations, 25th and 75th quantiles, and the 95% confidence interval. For AeroCom Phase II, the line indicates the mean \pm 1 sigma, and the minimum and maximum values from individual models are shown as crosses. (Figure 1d in the main text).

#2 AAOD to DRF:

This is discussed in Lines 218-227. I will review issues for BC only but they are true of the other absorbing species as well.

In model studies AAOD is found proportional to BC concentration and DRF is also proportional to BC concentration, thus AAOD and DRF may be assumed proportional. However this study estimates that AAOD is higher than ref 6 yet DRF is much lower, and

conversely AAOD is much higher (67%) than AEROCOM but forcing is similar. Therefore the AAOD to DRF ratio is different in this study than in previous studies.

Authors acknowledge this issue in these lines. Yet although this is the main novel finding of this manuscript (in my view), the only discussion is this: “the satellite products provide additional details in aerosol spatial distribution that are non-negligible for radiative effect estimates and hardly seen from the ground-based network.” If this is the case, then it is important and needs to be isolated and also given as a challenge to future modeling efforts! Has POLDER placed more absorbing aerosol below clouds where it has less forcing? Or in different regions or over ocean where there is less bright surface? A reduction of 50-70% in forcing efficiency that is constrained by spaceborne measurement would be quite notable, but it is not emphasized. Authors have all the data to investigate this question – model fields and observed fields.

Response:

Table R1 lists the BC DRF, Anthropogenic BC AAOD, BC AAOD, BC Anthropogenic Fraction and BC Radiative Forcing Efficiency (W/m^2 per unit of anthropogenic AAOD). Generally, our BC forcing efficiency (75 W/m^2) is lower than other studies about ~30% to ~100% (AeroCom Phase II 133 W/m^2 [84.3, 216]; MACv2 ~ 150 W/m^2 ; Bond2013 104 W/m^2). With respect to Heald et al. (2014) where the BC forcing efficiency is ~ 111 W/m^2 , we updated only the daily emissions in the GEOS-Chem simulation. Therefore, the spatial and temporal distribution of emission as well as the resulted simulation of aerosol absorption and its anthropogenic contribution should be responsible for the decrease of BC forcing efficiency.

There are several factors contributing to the forcing efficiency discrepancy. Some other studies also showed that the spatial and temporal distribution of anthropogenic AAOD and BC AAOD are crucial for further estimation of BC DRF (Reddy and Boucher, 2007; Räisänen et al., 2022). Basically, absorbing aerosol above dark surfaces (vegetation and ocean) will lead to less direct RF than over bright surfaces (desert, snow and ice). In addition, the effects over seasons also can be different due to different removal processes. We further calculate the regional BC radiative forcing efficiency showed in Figure R2. Interestingly, based on our simulation with MAP-constrained emissions, the BC forcing efficiency is much smaller (~ $40\text{-}70 \text{ W/m}^2$ per unit AAOD) over industrial regions than the other areas (105 W/m^2 per unit AAOD), which implies the future possible reduction of BC emissions over

industrial regions may be not efficient as expected for climate change mitigation (see also considerations in the answer #2). We've revised our manuscript with some discussions about forcing efficiency and its spatial variability.

Table R1. Intercomparison of BC DRF, Anthropogenic BC AAOD, BC AAOD, BC Anthropogenic Fraction and BC Radiative Forcing Efficiency.

	BC DRF (W/m ²)	Anthr. BC AAOD	Total BC AAOD	BC Anthr. Frac. (%)	BC Forcing Efficiency (W/m ²)
This study	+0.33	0.0044	0.0051	80.2	75
AeroCom PhaseII	+0.23	0.0015	-	-	133±37 [84.3, 216] *
MACv2	+0.28	-	-	50	~150
	+0.38	-	-	75	
	+0.45	-	-	85	
Bond (2013)	+0.51	0.0049	0.0061	80.8	104
Heald (2014)	+0.078	0.0007	-	-	111

* The range of minimum to maximum value from individual model.

	Anth. BC AAOD	BC RF Efficiency (W/m² per AAOD)	BC DRF (W/m²)
ESA	0.0183	57.4	1.05
SAS	0.0176	43.2	0.76
NAM	0.0092	39.1	0.36
EUR	0.0099	50.5	0.50
SAM	0.0070	44.3	0.31
CAF	0.0172	69.2	1.19
Rest of world	0.0019	105.3	0.20

Figure R2. The regional anthropogenic BC AAOD, BC DRF and BC DRF Efficiency based on our simulation.

Reddy, M. S. & Boucher, O. Climate impact of black carbon emitted from energy consumption in the world's regions. *Geophysical Research Letters* **34**, 11802 (2007).

Räisänen, P. *et al.* Mapping the dependence of BC radiative forcing on emission region and season. *Atmos. Chem. Phys. Discuss.* 1–31 (2022) doi:doi.org/10.5194/acp-2022-288.

This paper also mentions absorption cross section. A cross section is assumed that ignores internal mixing of particles which increases absorption. I assume that since the model is adjusted to match the AAOD, this choice doesn't matter: if the correct amount of absorption is in the atmosphere, the forcing should be approximately correct, even if the mass is wrong and the absorption per mass is wrong. This however is not stated.

Response:

Thanks very much for the constructive suggestions! Indeed, that's the reason why we focus mainly on the absorption and resulted DRF and less on the emissions. We now add this discussion in the revised manuscript.

#3 total radiative forcing:

This step requires a division of the total aerosol optical depth among other species. This is far more difficult than dividing among absorbing aerosol components (BC/OM/dust) which is not easy in itself. This division then propagates forward to the assumed optical properties of each species. As stated under #1, the optical properties (especially backscatter fraction) are key in determining negative forcing. I think this part should either be deleted, or it needs a far more expanded discussion of assumptions and uncertainties.

Response:

Thanks for pointing out this issue! We fully agree that our main focus is on the aerosol absorption. We also try to provide some limited suggestion for total aerosol effect (DRF). There are 2 main reasons for that: (i) - as stated in answers to question #3, we keep a standard scheme GEOS-Chem simulation of scattering aerosol species SNA and SS and obtain DRF for SNA (-0.39 W/m^2) and SS (0 W/m^2), which are in line with AEROCOM Phase II results (SNA: -0.40 W/m^2 ; SS: 0 W/m^2) (Myhre et al., 2013). Therefore, our estimation of scattering aerosol effects is reasonable from this point of view; (ii) - the total aerosol DRF is of high awareness and, for the completeness of the result presentation, it is good to display those number for allowing reader to compare them with other independent estimates. In addition, the major changes in total aerosol DRF are due to the updates of MAP-constrained absorbing species, since the estimation of scattering aerosol species are in agreement with current multi-model assessment. Due to all above reasons, we have decided to keep our discussion of total aerosol DRF in small space. Hopefully, you understand our decision.

Other comments

I suggest also comparison with Cohen and Wang, <https://doi.org/10.1002/2013JD019912>, which also used ground-based constraints to estimate BC emission.

Response:

We've added the result of Cohen and Wang (2014) in the revised manuscript:

“Our estimation of modern-era global BC emission is 18.4 Tg/yr, and 15.2 Tg/yr of that is emitted from anthropogenic origins. Our estimation of modern-era BC emission is consistent with previous estimate of 14.6 – 22.2 Tg/yr by Cohen and Wang (2014) using AERONET

AAOD as observational constraints, even though both of them are at least a factor of 2 higher than current bottom-up inventories.”.

I am puzzled by the title which describes only the portion that has already been published, i.e. authors' paper in 2019 covered emission and aerosol absorption.

Response:

Thanks! We decided to compress and refine our title to “Multi-angular polarimetric remote sensing to pinpoint global aerosol absorption”.

Reviewer #1 (Remarks to the Author):

I thank the authors for their thorough revision, and again for their highly relevant work on this subject. My concerns have been adequately addressed in the revision, and I therefore recommend that this manuscript be accepted for publication.

Reviewer #2 (Remarks to the Author):

I have summarized the noteworthy results and significance of this work in my original review. I believe that this work is valuable and I would like to see it published. The revision has addressed several of the points made by two reviewers. However, the revised paper also doesn't really answer some of the most important questions raised in review.

I am only reporting here on the issues that I feel were not well addressed in the revision. So, although these comments all appear negative, it is because I am putting my attention on the part where work remains. I appreciate authors' attention to many of the other comments. I recognize this is a complex topic and it is easy for reviewers to recommend investigations that go beyond the scope of a single paper. I hope I can strike a balance with suggestions that allow the authors to communicate the boundaries of their work while also not demanding too much extra work.

I start with the question of error bars. In the response, authors acknowledged the uncertainty was originally determined with the interannual fluctuation from 2001-2006 (this is variability, not uncertainty). They have now updated the uncertainty to include differences in anthropogenic fraction. They acknowledge that they have determined it with a single model and cannot do a multi-model estimate – I have no problem with this. But then, they should not present an overall uncertainty without acknowledging what they have not considered in that estimate – the statement should be something like "estimate is [X-Y], considering only effects 1,2,3 and not 4,5,6."

Then the issue of forcing efficiency. Authors have come up with a lower forcing efficiency than any model has previously produced, stating that the forcing efficiency is lower in industrial regions. I agree that forcing efficiency is spatially dependent. However, in the response (but not the paper), they state "Basically, absorbing aerosol above dark surfaces (vegetation and ocean) will lead to less direct RF than over bright surfaces (desert, snow and ice)." But this is (crudely) opposite of the observation: industrial regions, if land surface is modified, should also have brighter surfaces compared with ocean and forests. Authors also refer to Derimian et al 2016 to support their somewhat mystifying statement that "DRF is affected stronger by non-linear multiple scattering effects resulting in lower DRF efficiency." (line 635-636 in the red-line submission.) But Derimian et al. didn't explore absorbing aerosol in this way; all the comparisons in that paper are of mixed aerosol, not black carbon alone, which are of course, with a higher scattering fraction, more subject to multiple scattering. Furthermore, this doesn't explain why DRF has similar efficiency even when the AAOD is halved (Table 1).

****In short, this finding of lower DRF efficiency is possibly one of the more important findings of the paper, but it is not carefully explained: the arguments are not well supported and do not yet reflect a fact-based inquiry into the causes.****

This then leads to the question of effective radiative forcing, ERF, raised by reviewer 1. The authors don't use ERF as does IPCC-AR6, but rather direct radiative forcing which is consistent with prior practice. I accept this, but their justification added to the revised paper is: "Nonetheless the rapid adjustments strongly rely on the highly uncertain simulation of global aerosol absorption magnitude, especially its spatial distribution." (In other words these adjustments are too uncertain to be considered in this paper?) This argument isn't compelling because authors suggest that an advantage of their work

is the improved spatial distribution and aerosol absorption magnitude! Perhaps authors might like to suggest what type of information is needed to conduct a follow-on study with the present retrieval, acknowledging the importance of ERF. As an aside: I previously suggested the collocation of aerosol and cloud, and relative position (aerosol above or below cloud) as a possible reason for change in DRF. This would also affect ERF, which should then be revisited, with a separate study.

Title: Authors have focused the title to "Multi-angular polarimetric remote sensing to pinpoint global aerosol absorption." My main critique, as it was before, is that this title does not encapsulate the novel contribution of this paper. Authors already reported this value of AAOD in their earlier paper, "Constraining global aerosol emissions using POLDER/PARASOL satellite remote sensing observations" (ACP, 19, 14585–14606, 2019). I agree that the paper under review puts more emphasis on AAOD, but the value of AAOD is not its new contribution.

Finally the issue of total aerosol effect. I had previously a critique that the DRF of all aerosol (including e.g. sulfate in addition to BC) was not well explained nor uncertainties supported. In the rebuttal, authors provided more detail on their uncertainty exploration, also with the statements "We also try to provide some limited suggestion for total aerosol effect (DRF)." And "Due to all above reasons, we have decided to keep our discussion of total aerosol DRF in small space." What I interpret is that authors have indeed done a lot of work to understand the magnitude and uncertainties of total aerosol DRF, and this is valuable. But this exploration is minimized in the paper and thus a result for total aerosol DRF, is presented without the usual methodological rigor. I understand the space reasons, but the question is whether one should publish a direct forcing estimate and uncertainties, that will certainly be propagated forward to future assessments, without an explanation for the community about how that estimate was obtained. It is my feeling that this is a worthy investigation and should be in a separate paper so that it can be supported, but I leave that up to the editor.

Reviewer #1:

I thank the authors for their thorough revision, and again for their highly relevant work on this subject. My concerns have been adequately addressed in the revision, and I therefore recommend that this manuscript be accepted for publication.

Response:

Thanks very much for supports and overall constructive suggestions!

Reviewer #2:

I have summarized the noteworthy results and significance of this work in my original review. I believe that this work is valuable and I would like to see it published. The revision has addressed several of the points made by two reviewers. However, the revised paper also doesn't really answer some of the most important questions raised in review.

I am only reporting here on the issues that I feel were not well addressed in the revision. So, although these comments all appear negative, it is because I am putting my attention on the part where work remains. I appreciate authors' attention to many of the other comments. I recognize this is a complex topic and it is easy for reviewers to recommend investigations that go beyond the scope of a single paper. I hope I can strike a balance with suggestions that allow the authors to communicate the boundaries of their work while also not demanding too much extra work.

Response:

We fully agree that it is challenging to cover a complex topic including emission, absorption and radiative effects in a single paper and there are a lot things to clarify. Basically, we try to mention the entire work flow we realized to constrain emission of absorbing aerosol from MAP (PARASOL/GRASP) product and then use the updated emissions to simulate aerosol absorption and its radiative effects. By doing this, we would like to highlight one of the advantages of this study that is tight alignment (fusion) of models and observations. Overall, in our opinion, there are many aspects about amount of aerosol absorption and its global radiative effects that need to be studied. Therefore, there are a lot possibility for additional investigations and analyses. In this respect, we attempted to combine observation and model, by aligning tightly models with MAP products. Of course, we realize that this is one of the ways to move it forward that can't address and answer all questions. Moreover, it has some notable weaknesses that you have outline in your review. Therefore, we understand your concerns and justified critics that we are sure came with the intensions to strengthen to find

show arguments of our approach in most convincing way. Thank you very much for the constructive comments on our manuscript! We tried to revised/reshape our manuscript to show adequately and honestly both the strength and weaknesses of our approach and results. Please find our point-by-point response below.

I start with the question of error bars. In the response, authors acknowledged the uncertainty was originally determined with the interannual fluctuation from 2001-2006 (this is variability, not uncertainty). They have now updated the uncertainty to include differences in anthropogenic fraction. They acknowledge that they have determined it with a single model and cannot do a multi-model estimate – I have no problem with this. But then, they should not present an overall uncertainty without acknowledging what they have not considered in that estimate – the statement should be something like “estimate is [X-Y], considering only effects 1,2,3 and not 4,5,6.”

Response:

Thanks very much again for the very good suggestion! We’ve revised our text to mention that the uncertainty is largely derived from the interannual variability taking into account the anthropogenic fractions. There are many other factors that are not considered, for example, uncertainties of the model itself, chemical/physical processes, meteorology, spatial representativeness of satellite observation, limited spatial resolution of model simulation etc.

L257: *“Our aerosol DRF uncertainty range is driven by single model 6 years variability and perturbation of BC anthropogenic fractions. Many other factors, for example uncertainties of the model itself, the uncertainties in parameterization of chemical and physical processes, meteorology, limited spatial resolution of model simulation, etc., were not taken into account in the present estimation.”*

L267: *“It should be noted that the estimations of uncertainty range are different from study to study.”*

Then the issue of forcing efficiency. Authors have come up with a lower forcing efficiency than any model has previously produced, stating that the forcing efficiency is lower in industrial regions. I agree that forcing efficiency is spatially dependent. However, in the response (but not the paper), they state “Basically, absorbing aerosol above dark surfaces (vegetation and ocean) will lead to less direct RF than over bright surfaces (desert, snow and ice).” But this is (crudely) opposite of the observation: industrial regions, if land surface is

modified, should also have brighter surfaces compared with ocean and forests. Authors also refer to Derimian et al 2016 to support their somewhat mystifying statement that “DRF is affected stronger by non-linear multiple scattering effects resulting in lower DRF efficiency.” (line 635-636 in the red-line submission.) But Derimian et al. didn’t explore absorbing aerosol in this way; all the comparisons in that paper are of mixed aerosol, not black carbon alone, which are of course, with a higher scattering fraction, more subject to multiple scattering. Furthermore, this doesn’t explain why DRF has similar efficiency even when the AAOD is halved (Table 1).

In short, this finding of lower DRF efficiency is possibly one of the more important findings of the paper, but it is not carefully explained: the arguments are not well supported and do not yet reflect a fact-based inquiry into the causes.

Response:

Thanks a lot for pointing out this issue. We appreciate the reviewer’s suggestion to check forcing efficiency from beginning and help us to identify the interesting outcome with a lower forcing efficiency over industrial regions. Indeed, we do not dig in deeply in investigations this phenomenon because we may need a separate and comprehensive study in future. Here we propose several hypotheses and synthesis of them:

(i) Over industrial regions, there are larger fractions of absorbing aerosol below clouds than in other regions and that may offset its warming effects. Also, the effect can be associated with high cloud fraction and low absorbing aerosol layer. We have added this suggestion in the revised manuscript.

(ii) Non-linear multiple scattering effects resulting in lower DRF efficiency over industrial regions where the aerosol abundances are generally higher than other regions therefore a relative strong multiple scattering effect. In this respect, the DRF efficiency saturates and decrease once the multiple scattering regime dominates. This is one potential reason. We agree that the multiple scattering effects on BC alone may be not as obvious as other species with higher scattering fraction. At the same time, BC aerosols almost never dominate ambient aerosol, but rather present mixed with other aerosols. In this respect, we would like to emphasize Derimian et al. (2016) had as one of the main objectives to consider all aerosol situations appearing in practice including the domination of absorbing aerosols. With that purpose Derimian et al. (2016) used AERONET climatology for selecting the set of aerosol scenarios. For example, it used not new but very relevant and concise studies by Dubovik et al. (2002). At the same time, it is possible to state that AERONET data has become as one of most trusted source of information about absorption of ambient aerosol that widely used of

most solid sources of relevant information for critical analyzes of aerosol effects on climate, e.g. in IPCC reports.

(iii) The surface albedo is another factor. Basically, absorbing aerosol above dark surfaces (vegetation and ocean) will lead to less direct RF than over bright surfaces (desert, snow and ice). Indeed, the industrial regions usually have slightly higher albedo than dark ocean and dense vegetation covered regions. While it's still darker than the desert-, snow- or ice-covered areas. The 6 selected regions cover mainly industrial regions and its surrounded vegetation areas. Therefore, we would consider surface albedo as another important and uncertain factor.

Overall, we propose several potential reasons that are not fully considered in this study and deserve further investigation, by detailed analysis and extensive simulations.

This then leads to the question of effective radiative forcing, ERF, raised by reviewer 1. The authors don't use ERF as does IPCC-AR6, but rather direct radiative forcing which is consistent with prior practice. I accept this, but their justification added to the revised paper is: "Nonetheless the rapid adjustments strongly rely on the highly uncertain simulation of global aerosol absorption magnitude, especially its spatial distribution." (In other words these adjustments are too uncertain to be considered in this paper?) This argument isn't compelling because authors suggest that an advantage of their work is the improved spatial distribution and aerosol absorption magnitude! Perhaps authors might like to suggest what type of information is needed to conduct a follow-on study with the present retrieval, acknowledging the importance of ERF.

Response:

Thanks for pointing out this issue! We agree the importance of ERF and its accurate measure of the Earth's radiative imbalance as well as better indication of forcing on global temperature change. We have revised the sentence to:

"Nonetheless the heterogeneity of global spatial distribution of aerosol absorption has also certainly an impact on the rapid adjustments. Thus, the improved quantification of global spatial heterogeneity of aerosol absorption distribution is still highly demanded."

As an aside: I previously suggested the collocation of aerosol and cloud, and relative position (aerosol above or below cloud) as a possible reason for change in DRF. This would also affect ERF, which should then be revisited, with a separate study.

Response:

Thanks for the suggestion! We have added the discussion about aerosol above/below cloud as a potential reason of low DRF efficiency over industrial region. We also agree that it would be interesting to quantify the impact on the rapid adjustments and ERF by using the constrained aerosol absorption. Definitely, we will try to report on this in future studies.

Title: Authors have focused the title to “Multi-angular polarimetric remote sensing to pinpoint global aerosol absorption.” My main critique, as it was before, is that this title does not encapsulate the novel contribution of this paper. Authors already reported this value of AAOD in their earlier paper, “Constraining global aerosol emissions using POLDER/PARASOL satellite remote sensing observations” (ACP, 19, 14585–14606, 2019). I agree that the paper under review puts more emphasis on AAOD, but the value of AAOD is not its new contribution.

Response:

Indeed, the emission and global aerosol absorption for year 2010 have been reported in our previous studies (Chen et al., 2018, 2019). Here, we extend our method to 6 years (2006-2011) and estimate the emission, aerosol absorption and radiative effects’ contribution from anthropogenic emissions. We have modified the title to “Multi-angular remote sensing to pinpoint global aerosol absorption and direct radiative forcing” in order to highlight the importance of multi-angular polarimetric remote sensing for aerosol absorption study and open many possibilities with the future missions (3MI, CO2M, PACE, etc.).

Finally the issue of total aerosol effect. I had previously a critique that the DRF of all aerosol (including e.g. sulfate in addition to BC) was not well explained nor uncertainties supported. In the rebuttal, authors provided more detail on their uncertainty exploration, also with the statements “We also try to provide some limited suggestion for total aerosol effect (DRF).” And “Due to all above reasons, we have decided to keep our discussion of total aerosol DRF in small space.” What I interpret is that authors have indeed done a lot of work to understand the magnitude and uncertainties of total aerosol DRF, and this is valuable. But this exploration is minimized in the paper and thus a result for total aerosol DRF, is presented without the usual methodological rigor. I understand the space reasons, but the question is whether one should publish a direct forcing estimate and uncertainties, that will certainly be propagated forward to future assessments, without an explanation for the community about how that estimate was obtained. It is my feeling that this is a worthy investigation and should be in a separate paper so that it can be supported, but I leave that up to the editor.

Response:

Thank you very much for the suggestion! Indeed, we constrained 3 absorbing aerosol species (BC, OC and DD). The reason why we intend to report all aerosol DRF including scattering sulfate is to check the overall effects from our constrained aerosol absorption and to intercompare with other estimations (e.g. IPCC and AEROCOM). Meanwhile, we use the GEOS-Chem RRTMG v11-01 simulation that is documented in Heald et al. (2015) and obtain the DRF -0.39 W/m^2 for sulfate-nitrate-ammonium (SNA) which is in line with IPCC and AEROCOM $\sim -0.4 \text{ W/m}^2$. Hence, the changes of DRF from our estimation is mainly driven by constrained absorbing aerosols. We agree with the reviewer that our estimated aerosol DRF uncertainty is less supportable. Our aerosol DRF uncertainty is estimated based on the variation of BC anthropogenic fraction together with the annual variability of other aerosol species. We have revised the manuscript to include a description of our uncertainty range.

L257: *“The reported aerosol DRF uncertainty range is driven by single model 6 years variability and perturbation of BC anthropogenic fractions. Many other factors, for example uncertainties of the model itself. For example, the uncertainties in parameterization of chemical and physical processes, meteorology, limited spatial resolution of model simulation, etc., were not taken into account in the present estimation.”*